# On the Computational Complexity of Performative Prediction

**Ioannis Anagnostides** [*1] **Rohan Chauhan** [2] **Ioannis Panageas** [2] **Tuomas Sandholm** [1 3] **Jingming Yan** [*2]

## Abstract

Performative prediction captures the phenomenon where deploying a predictive model shifts the underlying data distribution. While simple retraining dynamics are known to converge linearly when the performative effects are weak ($\rho < 1$), the complexity in the regime $\rho > 1$ was hitherto open. In this paper, we establish a sharp phase transition: computing an $\epsilon$-performatively stable point is PPAD-complete—and thus polynomial-time equivalent to Nash equilibria in general-sum games—even when $\rho = 1 + O(\epsilon)$. This intractability persists even in the ostensibly simple setting with a quadratic loss function and linear distribution shifts. One of our key technical contributions is to extend this PPAD-hardness result to general convex domains, which is of broader interest in the complexity of variational inequalities. Finally, we address the special case of strategic classification, showing that computing a strategic local optimum is PLS-hard.

## 1. Introduction

Machine learning models are typically developed under the assumption that the deployment environment is static: the underlying data distribution remains fixed regardless of the model's predictions. However, in many high-stakes social and economic domains, this premise is fundamentally flawed. As sociologists and economists have long observed, models are not merely cameras that passively record the markets, but engines that actively shape the reality they aim to model (MacKenzie, 2008). Similarly, in modern predictive tasks—ranging from credit scoring to spam filtering—the deployment of a predictive model triggers a shift in the underlying data distribution, as agents react strategically to the deployed classifier.

This ubiquitous phenomenon was formalized by Perdomo et al. (2020) as *performative prediction*. In this setting, the model, parameterized by $\boldsymbol{x} \in \mathcal{X}$, induces a distribution $\mathcal{D}(\boldsymbol{x})$ over the data. The decision-maker is thus facing a moving target: updating the model triggers a shift in the very objective they seek to minimize. This feedback loop yields two natural solution concepts. First, a *performatively optimal* point minimizes the expected loss over the distribution it induces, $\mathbb{E}_{\boldsymbol{z} \sim \mathcal{D}(\boldsymbol{x})}[\ell(\boldsymbol{x}; \boldsymbol{z})]$ (Definition 2.1). In contrast, a *performatively stable* point is based on a fixed-point consideration: it induces a distribution and simultaneously minimizes loss for that distribution (Definition 2.2).

Stability is a key desideratum, ensuring that the model remains invariant under retraining. Perhaps the most natural algorithmic approach to finding such points is by *repeatedly* solving the risk minimization problem—a process coined *repeated risk minimization (RRM)* by Perdomo et al. (2020)—until a fixed point is reached.

**Definition 1.1** (Perdomo et al., 2020). *Repeated risk minimization (RRM)* refers to the procedure whereby, starting from an initial model parameterized by $\boldsymbol{x}_0$, the following sequence of updates is performed.

$$\boldsymbol{x}_{t+1} = G(\boldsymbol{x}_t) = \operatorname*{argmin}_{\boldsymbol{x} \in \mathcal{X}} \mathbb{E}_{\boldsymbol{z} \sim \mathcal{D}(\boldsymbol{x}_t)}[\ell(\boldsymbol{x}; \boldsymbol{z})].$$

Perdomo et al. (2020) showed that RRM is bound to converge linearly to a performatively stable point when $\rho := L\beta/\alpha < 1$. Intuitively, this condition requires that the sensitivity of the distribution shift $L$ is small relative to the geometry of the loss landscape, governed by its smoothness $\beta$ and strong convexity $\alpha$ (Section 2 contains the precise definitions). On the other hand, Perdomo et al. (2020) observed that RRM can fail to converge even when $\rho = 1$, which means that performative effects are marginally stronger.

The condition $\rho < 1$ is quite restrictive. For example, without strong convexity the parameter $\rho$ can be unbounded; in many practical applications, the loss function is not even convex, let alone strongly convex (Li & Wai, 2024). Indeed, the regime where $\rho > 1$ is prevalent and arises even in simple problems, as pointed out by Perdomo et al. (2020) in their experimental simulations.

One might hope to circumvent this by simply increasing

[1]Carnegie Mellon University [2]University of California, Irvine [3]Strategy Robot, Inc., Strategic Machine, Inc., Optimized Markets, Inc.. Correspondence to: <ianagnos@cs.cmu.edu, jingmy1@uci.edu>.

*Proceedings of the 43rd International Conference on Machine Learning*, Seoul, South Korea. PMLR 306, 2026. Copyright 2026 by the author(s).

the regularization strength $\alpha$ to force the condition $\rho < 1$. However, such additional regularization can be destructive, potentially eliminating the meaningful equilibria.

Despite the significant progress in establishing improved convergence guarantees (*e.g.*, Khorsandi et al., 2025; Mofakhami et al., 2023), the complexity of computing performatively stable points remains poorly understood when $\rho \geq 1$. In particular, a fundamental question arises:

> *Is the failure of RRM simply a limitation of specific retraining dynamics, or is identifying a performatively stable point intrinsically intractable in the presence of stronger performative effects?*

The failure of RRM—and other algorithms such as repeated gradient descent and performative gradient descent (Izzo et al., 2021)—when $\rho \geq 1$ does not by itself imply intractability. Indeed, as we show, there are efficient algorithms even when $\rho$ is slightly above 1.

## 1.1. Our results

We characterize the computational complexity of performative stability across the spectrum of $\rho$. To begin with, we establish the following result.

**Theorem 1.2.** *For any small enough $\epsilon > 0$, computing an $\epsilon$-performatively stable point for some $\rho = L\beta/\alpha \leq 1 + O(\epsilon)$ is PPAD-hard.*

This means that computing performatively stable points is as hard as finding Nash equilibria in general-sum games (Daskalakis et al., 2009; Chen et al., 2009), which is unlikely to admit efficient algorithms. There is a basic trade-off in Theorem 1.2 worth highlighting: PPAD-hardness persists even if one is content with a crude approximation $\epsilon = \Theta(1)$, but that only precludes instances in which $L\beta/\alpha$ is some additive constant larger than 1. At the other end of the spectrum, PPAD-hardness kicks in even when $L\beta/\alpha - 1$ is exponentially small, as long as the desired precision is also small enough.

We also show that an $\epsilon$-performatively stable point can be computed in $\text{poly}(d, \log(1/\epsilon))$ time when $\rho = 1 + O_\epsilon(\epsilon^4)$ (Theorem 3.5). This improves upon repeated risk minimization (RRM) and other natural algorithms. While RRM converges for $\rho < 1$, the number of iterations scales with $\log^{-1}(1/\rho) \approx 1/1-\rho$ when $\rho \approx 1$, which blows up when $\rho$ approaches 1. Moreover, a recent result by Diakonikolas (2025) implies a $\text{poly}(1/\epsilon)$ algorithm when $\rho \leq 1 + O_\epsilon(\epsilon)$, matching Theorem 1.2 in the order of $\epsilon$. As a result, we find the transition from PPAD-hardness to tractability to be particularly acute.

Furthermore, we establish unconditional, information-theoretic lower bounds, showing that any algorithm requires exponentially many *empirical risk minimization (ERM)* evaluations to find a performatively stable point (Corollary 3.7). This holds whether one uses repeated risk minimization or any other more sophisticated algorithm.

From a technical standpoint, our hardness results are established through simple, canonical reductions that encode any variational inequality or fixed point problem as an instance of performative stability (Propositions 3.1 and 3.2). In particular, we show that intractability persists even in the ostensibly simple setting comprising a quadratic loss and an affine distribution shift (Theorem 3.4).

**The nonexpansive regime**   We go on to generalize the setup of performative prediction to general norms, extending the $\ell_2$ contraction argument of Perdomo et al. (2020) (Section 3.2). Interestingly, we observe that in this generalized setting, intractability barriers emerge *even in the contractive regime*. Specifically, we show that finding performatively stable points would imply a complexity theory breakthrough (Proposition 3.10).

**The role of the constraint set**   An important component of our reduction is the geometry of the domain. Existing hardness results for variational inequalities and fixed points typically rely on the hypercube $\mathcal{X} = [0, 1]^d$. However, this does not always mesh well with machine learning applications; for example, in the context of performative prediction, training a classifier constrained on the $\ell_2$ ball instead is perhaps more natural (Hinton et al., 2012; Goodfellow et al., 2016). Surprisingly, the complexity of VIs and fixed points over general constraint sets has received limited attention, with some exceptions (Section 1.2). We fill this gap by showing that PPAD-hardness persists under any reasonable convex constraint set (Theorem 3.12).

**Strategic classification**   Finally, we turn to *strategic classification* (Hardt et al., 2016), which falls within the scope of performative prediction. We show that finding a *local* optimum of the performative risk—under single-label updates—is PLS-hard (Theorem 4.4); PLS captures the complexity of (presumably) hard local optimization problems such as local max-cut. This complements the NP-hardness of Hardt et al. (2016) concerning *global* performative optimality, and further highlights the intractability of attaining performative optimality. It shows that local search heuristics—often employed to sidestep NP-hardness—can fail to efficiently identify stable points.

## 1.2. Related work

Following the foundational work of Perdomo et al. (2020), significant attention has been devoted to the convergence of retraining dynamics. Mendler-Dünner et al. (2020)

analyzed stochastic variants of RRM, distinguishing between "greedy" and "lazy" deployment. In their terminology, greedy deployment releases the new model at every step, whereas lazy deployment accumulates multiple gradient updates before releasing a new model. Zrnic et al. (2021) further refined those dynamics by studying two-timescale algorithms, showing that separating the timescales of model updates and the resulting distribution shifts can stabilize learning. Miller et al. (2021) and Izzo et al. (2021) developed derivative-free methods to optimize the performative risk, targeting optimality rather than just stability. For more recent pointers, we refer to Khorsandi et al. (2025); Mofakhami et al. (2023); Brown et al. (2022), and references therein. There has been some work addressing misspecification in the underlying distribution map (Xue & Sun, 2024), but our paper operates in the standard model. (Of course, our lower bounds readily apply under such misspecification.)

**Strategic classification**  Performative prediction encompasses the framework of *strategic classification* (Hardt et al., 2016; Chen et al., 2018; 2020; Dong et al., 2018; Meir et al., 2012), where the distribution shift arises from individual agents rationally best-responding to the classifier. The performative prediction framework abstracts the game-theoretic interaction into the distribution map.

**Multiagent performative prediction**  While the original framework of Perdomo et al. (2020) considers a single decision-maker, many real-world applications involve multiple agents with different objectives. In such settings, the underlying distribution can depend on the joint deployment of all agents, giving rise to more complex distribution shifts. Piliouras & Yu (2022) formalized this as *multiagent performative prediction*, and showed that standard retraining dynamics can lead to complex behaviors, ranging from global stability to chaos. Performative prediction in multi-player settings has since received considerable attention (Narang et al., 2023; Góis et al., 2025). Although we draw upon techniques from algorithmic game theory, our results pertain to the single-agent setting.

Drusvyatskiy & Xiao (2023) analyze performative stability through the lens of stochastic optimization with decision-dependent distributions, applying proximal-point methods and extra-gradient algorithms in performative prediction.

**Performative prediction and reinforcement learning**  Dynamic feedback effects similar to performative prediction are also common in reinforcement learning and robotics, where the act of learning can change the environment (Levine et al., 2020). This connection has motivated a line of research on performative stability in reinforcement learning. In particular, Mandal et al. (2023) studied the no-

tion of a performatively stable policy in single-agent reinforcement learning, while Rank et al. (2024) extended this notion to settings in which the learning environment itself can shift in response to the deployment of different policies. Furthermore, Basu et al. (2025) proposed a policy-gradient algorithm that converges to performatively optimal policies.

**Complexity theory**  PPAD was introduced by Papadimitriou (1994) and was famously shown to characterize the complexity of Nash equilibria in two-player general-sum games (Daskalakis et al., 2009; Chen et al., 2009). Our PPAD-hardness proof for general convex sets leverages certain tools developed by Mehta (2014) and Deligkas et al. (2020). As we highlighted earlier, much of the work in the complexity of fixed points and variational inequalities has focused on the case where the constraint set is the hypercube. A notable recent exception is the paper of Attias et al. (2025), which proves exponential lower bounds when the constraint set is the $\ell_2$ ball.

PLS was introduced by Schäffer & Yannakakis (1991) to characterize the complexity of (presumably) hard local optimization problems, such as max-cut under the so-called flip neighborhood. It is also known to characterize the complexity of *pure* Nash equilibria in multi-player potential games (Fabrikant et al., 2004). As a result, our results imply a polynomial-time equivalence between the complexity of local performative optimality and pure Nash equilibria in potential games.

For a comprehensive overview of the emerging field of performative prediction, we refer to Hardt & Mendler-Dünner (2025).

## 2. Preliminaries

**Notation**  For $x, x' \in \mathbb{R}^d$, we use $\langle x, x' \rangle$ for their inner product. $\|x\|_2 = \sqrt{\langle x, x \rangle}$ denotes the Euclidean norm. $\|\cdot\|$ denotes an arbitrary norm. $\mathcal{X}$ is a convex and compact subset of $\mathbb{R}^d$ that represents the parameter space of the decision-maker. For $x \in \mathcal{X}$, $\mathcal{D}(x)$ denotes the distribution induced by $x$. We will use the notation $\mathcal{Z} := \bigcup_{x \in \mathcal{X}} \text{supp}(\mathcal{D}(x))$. For the sake of exposition, we sometimes write $O_\epsilon(\cdot)$ to denote the dependence only on the parameter $\epsilon$.

Performative prediction centers on the problem

$$\min_{x \in \mathcal{X}} \mathbb{E}_{z \sim \mathcal{D}(x)} \left[ \ell(x; z) \right]. \tag{1}$$

We now formally define performative optimality and performative stability.

**Definition 2.1** (Performative optimality; Perdomo et al.,

2020). A point $\boldsymbol{x}^* \in \mathcal{X}$ is *performatively optimal* if

$$\boldsymbol{x}^* \in \underset{\boldsymbol{x} \in \mathcal{X}}{\operatorname{argmin}} \, \mathbb{E}_{\boldsymbol{z} \sim \mathcal{D}(\boldsymbol{x})} \left[ \ell(\boldsymbol{x}; \boldsymbol{z}) \right].$$

Performatively optimal points correspond to *Stackelberg equilibria* (Conitzer & Sandholm, 2006; von Stackelberg, 1934), as the decision-maker commits to a model, anticipating how the distribution $\mathcal{D}$ will shift in response. In Section 4, we introduce a local version of Definition 2.1 in the context of strategic classification.

**Definition 2.2** (Performative stability; Perdomo et al., 2020). A point $\boldsymbol{x}^* \in \mathcal{X}$ is *performatively stable* if

$$\boldsymbol{x}^* \in \underset{\boldsymbol{x} \in \mathcal{X}}{\operatorname{argmin}} \, \mathbb{E}_{\boldsymbol{z} \sim \mathcal{D}(\boldsymbol{x}^*)} \left[ \ell(\boldsymbol{x}; \boldsymbol{z}) \right].$$

Performatively stable points exist under mild assumptions (Perdomo et al., 2020). They are in correspondence to *Nash equilibria*, as the decision-maker selects a model that is optimal for the *current* distribution. Our complexity results leverage this connection to shed light on the complexity of performative prediction.

The following assumptions are made concerning the loss function and the magnitude of the distribution shift.

**Assumption 2.3.** *Let $\ell(\boldsymbol{x}; \boldsymbol{z})$ be the loss function and $\mathcal{D}(\boldsymbol{x})$ the distribution on $\mathcal{Z}$ induced by $\boldsymbol{x} \in \mathcal{X}$.*

- *(strong convexity) $\ell(\boldsymbol{x}; \boldsymbol{z})$ is $\alpha$-strongly convex with respect to $\| \cdot \|_2$:*

$$\ell(\boldsymbol{x}; \boldsymbol{z}) \geq \ell(\boldsymbol{x}'; \boldsymbol{z}) + \langle \nabla_{\boldsymbol{x}} \ell(\boldsymbol{x}'; \boldsymbol{z}), \boldsymbol{x} - \boldsymbol{x}' \rangle + \frac{\alpha}{2} \| \boldsymbol{x} - \boldsymbol{x}' \|_2^2$$

  *for any $\boldsymbol{x}, \boldsymbol{x}' \in \mathcal{X}$ and $\boldsymbol{z} \in \mathcal{Z}$.*
- *(smoothness) $\ell(\boldsymbol{x}; \boldsymbol{z})$ is $\beta$(-jointly) smooth if*

$$\| \nabla_{\boldsymbol{x}} \ell(\boldsymbol{x}; \boldsymbol{z}) - \nabla_{\boldsymbol{x}} \ell(\boldsymbol{x}'; \boldsymbol{z}) \|_2 \leq \beta \| \boldsymbol{x} - \boldsymbol{x}' \|_2$$

  *and*

$$\| \nabla_{\boldsymbol{x}} \ell(\boldsymbol{x}; \boldsymbol{z}) - \nabla_{\boldsymbol{x}} \ell(\boldsymbol{x}; \boldsymbol{z}') \|_2 \leq \beta \| \boldsymbol{z} - \boldsymbol{z}' \|_2$$

  *for any $\boldsymbol{x}, \boldsymbol{x}' \in \mathcal{X}$ and $\boldsymbol{z}, \boldsymbol{z}' \in \mathcal{Z}$.*
- *(sensitivity) $\mathcal{D}$ is $L$-sensitive if*

$$W_1(\mathcal{D}(\boldsymbol{x}), \mathcal{D}(\boldsymbol{x}')) \leq L \| \boldsymbol{x} - \boldsymbol{x}' \|_2$$

  *for any $\boldsymbol{x}, \boldsymbol{x}' \in \mathcal{X}$, where $W_1$ denotes the Wasserstein-1 distance, or earth mover's distance.*

We define $\rho := L\beta/\alpha$. In Section 3.2, we also generalize the setup of Assumption 2.3 to general norms.

We rely on the following notion of approximation.

**Definition 2.4.** A point $\boldsymbol{x}^* \in \mathcal{X}$ is (first-order) $\epsilon$-*performatively stable* if

$$\langle \boldsymbol{x} - \boldsymbol{x}^*, \mathbb{E}_{\boldsymbol{z} \sim \mathcal{D}(\boldsymbol{x}^*)} \left[ \nabla_{\boldsymbol{x}} \ell(\boldsymbol{x}^*; \boldsymbol{z}) \right] \rangle \geq -\epsilon \quad \forall \boldsymbol{x} \in \mathcal{X}.$$

When the loss function is convex, the definition above coincides with Definition 2.2 (Claim A.3). In applications where the loss function is nonconvex (Li & Wai, 2024), Definition 2.4 is the natural local relaxation of Definition 2.2. Since computing (first-order) performatively stable points lies in PPAD (Corollary A.2), our hardness result establishes PPAD-completeness for that notion.

Another natural way to measure the approximation error is through the fixed point gap $\| \boldsymbol{x}^* - G(\boldsymbol{x}^*) \|_2$, where $G$ is the RRM map (Definition 1.1); as we formalize in Lemmas F.1 and F.2, those notions are polynomially related.

# 3. Complexity of performatively stable points

In this section, we characterize the complexity of performatively stable points.

**A hard class of problems** We consider the following class of performative prediction instances.

$$\min_{\boldsymbol{x} \in \mathcal{X}} \left\{ \ell(\boldsymbol{x}; \boldsymbol{z}) := \frac{1}{2} \| \boldsymbol{x} \|_2^2 - \boldsymbol{x}^\top \boldsymbol{z} \right\}, \tag{2}$$

$$\text{where } \boldsymbol{z} = g(\boldsymbol{x}). \tag{3}$$

We assume that $g$ is $L$-Lipschitz continuous, so that $\| g(\boldsymbol{x}) - g(\boldsymbol{x}') \|_2 \leq L \| \boldsymbol{x} - \boldsymbol{x}' \|_2$ for any $\boldsymbol{x}, \boldsymbol{x}' \in \mathcal{X}$. The function $\ell$ defined in (2) is 1-strongly convex in $\boldsymbol{x}$ and 1-jointly smooth (per Assumption 2.3), while the sensitivity of $\mathcal{D}(\boldsymbol{x})$ is $L$. So, $\rho = L$ in this class.

The underlying distribution above is a singleton supported on $g(\boldsymbol{x})$. (The Wasserstein-1 distance between two point mass distributions is simply the distance between the two points.) Our reductions work more broadly for any distribution $\mathcal{D}(\boldsymbol{x})$ such that $\mathbb{E}_{\boldsymbol{z} \sim \mathcal{D}(\boldsymbol{x})} \boldsymbol{z} = g(\boldsymbol{x})$; this could be, for example, a more well-behaved Gaussian distribution. This holds because $\mathbb{E}_{\boldsymbol{z} \sim \mathcal{D}(\boldsymbol{x})}[\ell(\boldsymbol{x}; \boldsymbol{z})] = \mathbb{E}_{\boldsymbol{z} \sim \mathcal{D}(\boldsymbol{x})}[\frac{1}{2} \| \boldsymbol{x} \|^2 - \boldsymbol{x}^\top \boldsymbol{z}] = \frac{1}{2} \| \boldsymbol{x} \|^2 - \boldsymbol{x}^\top \mathbb{E}_{\boldsymbol{z} \sim \mathcal{D}(\boldsymbol{x})}[\boldsymbol{z}]$, by definition of the loss $\ell$ in (1). In other words, the choice of distribution does not alleviate the hardness of the problem.

## 3.1. Encoding VIs and fixed points

We now show how a suitable choice of $g$ allows us to encode hard optimization problems. First, we consider a *variational inequality (VI)* problem given by a mapping $F : \mathcal{X} \to \mathbb{R}^d$. An $\epsilon$-approximate VI solution is a point $\boldsymbol{x}^* \in \mathcal{X}$ such that $\langle \boldsymbol{x} - \boldsymbol{x}^*, F(\boldsymbol{x}^*) \rangle \geq -\epsilon$ for all $\boldsymbol{x} \in \mathcal{X}$. We observe that by selecting $g : \boldsymbol{x} \mapsto \boldsymbol{x} - F(\boldsymbol{x})$,

an $\epsilon$-performatively stable point of (2)-(3) yields an $\epsilon$-approximate solution to the VI problem. Furthermore, considering $g : \boldsymbol{x} \mapsto \boldsymbol{x} - \overline{F}(\boldsymbol{x})$ for a damped (rescaled) mapping $\overline{F}$ makes the Lipschitz constant of $g$—the sensitivity of the distribution—approach 1 while rescaling the approximation factors between the two problems. We summarize this guarantee below.

**Proposition 3.1** (From VIs to performative stability). *For any $\epsilon > 0$ and $\epsilon' > 0$, computing an $\epsilon'$-approximate VI solution of an $L$-Lipschitz mapping $F$ reduces to computing $\epsilon$-performatively stable points with $\rho \leq 1 + \frac{\epsilon}{\epsilon'}L$.*

We instantiate and sharpen this reduction in Theorem 3.4 for the class of affine VI problems. First, we provide a similar reduction for fixed point problems. Here, we are given a continuous function $T : \mathcal{X} \to \mathcal{X}$ and the problem is to find an $\epsilon$-fixed point thereof. We observe that by selecting $g : \boldsymbol{x} \mapsto (1 - \lambda)\boldsymbol{x} + \lambda T(\boldsymbol{x})$ for $\lambda = \epsilon/\epsilon'$, we arrive at the following theorem.

**Proposition 3.2** (From fixed points to performative stability). *For any $\epsilon > 0$, computing an $\epsilon'$-fixed point of an $L$-Lipschitz continuous mapping reduces to computing an $\epsilon$-fixed point of the RRM map $G$ (Definition 1.1) with $\rho \leq 1 + \frac{\epsilon}{\epsilon'}L$.*

It is well-known that fixed points can be reduced to VIs and vice versa, but the reduction above is particularly direct. We will use it to prove query lower bounds through the result of Hirsch et al. (1989).

We now leverage Proposition 3.1 to establish PPAD-hardness for a particularly simple class of problems: the distribution shift is given by an affine function and the constraint set $\mathcal{X}$ is the hypercube. We begin by extracting a useful result from Bernasconi et al. (2024, Theorem 4.4) concerning the complexity of affine VIs on the hypercube. A closely related result was shown by Rubinstein (2015) in the context of computing approximate Nash equilibria in polymatrix, binary-action games. For our purposes, it is convenient to use the lemma as given by Bernasconi et al. (2024) because of the assumed matrix bounds. For a matrix $\mathbf{A} \in \mathbb{R}^{d \times d}$, we denote by $\|\mathbf{A}\|_1$ its maximum absolute column sum and by $\|\mathbf{A}\|_\infty$ its maximum absolute row sum. Our proof uses the fact that the spectral norm $\|\mathbf{A}\|_2$ satisfies the inequality $\|\mathbf{A}\|_2 \leq \sqrt{\|\mathbf{A}\|_1 \|\mathbf{A}\|_\infty}$.

**Lemma 3.3** (Bernasconi et al., 2024). *It is PPAD-complete to find a point $\boldsymbol{x}^* \in [0, 1]^d$ such that for all $\boldsymbol{x} \in [0, 1]^d$,*

$$\langle \boldsymbol{x} - \boldsymbol{x}^*, \mathbf{A}\boldsymbol{x}^* + \boldsymbol{b} \rangle \geq -\epsilon'.$$

*This holds even when $\epsilon' > 0$ is an absolute constant, $\|\mathbf{A}\|_1 \leq 1$, and $\|\mathbf{A}\|_\infty \leq 1$.*

This allows us to strengthen Proposition 3.1 by establishing PPAD-hardness with sharp constants and for a seemingly simple class of performative prediction instances.

**Theorem 3.4.** *Finding an $\epsilon$-performatively stable point per Definition 2.4 is PPAD-hard even when $L\beta/\alpha \leq 1 + \frac{\epsilon}{\epsilon'}$ for $\epsilon' = 0.088/6 \approx 0.0147$. This is so even when $\ell$ is a quadratic objective, $\ell(\boldsymbol{x}; \boldsymbol{z}) = \frac{1}{2}\|\boldsymbol{x} - \boldsymbol{z}\|_2^2$, and $\mathcal{D}(\boldsymbol{x})$ is given by an affine map.*

The simple proof is deferred to Appendix F. The constant $\epsilon'$ appearing above is obtained by combining Lemma 3.3 with the inapproximability of Deligkas et al. (2024). It is worth noting that for general VIs on the hypercube (without the restriction to affine mappings), an approximation of even $\approx \frac{1}{2}$ is PPAD-hard (Deligkas et al., 2023).

To put Theorem 3.4 into context, Diakonikolas (2025) recently analyzed a relaxation of nonexpansiveness, showing that an $\epsilon$-fixed point—which yields an $O_\epsilon(\epsilon)$-performatively stable point on account of Lemma F.2—can be computed in $\text{poly}(1/\epsilon)$ time even when the Lipschitz constant $L$ of the map satisfies $L \leq 1 + \epsilon/D$. This means that, subject to a complexity collapse, the bound on $\rho$ in Theorem 3.4 cannot be improved up to the factor multiplying $\epsilon$. Taken together, we identify an acute phase transition in the complexity of the problem.

Furthermore, we also establish the following result.

**Theorem 3.5.** *If $\rho \leq 1 + \epsilon$ (per Assumption 2.3), there is a $\text{poly}(d, \log(1/\epsilon))$-time algorithm for computing an $O_\epsilon(\epsilon^{1/4})$-performatively stable point.*

The approximation we obtain holds even in terms of the RRM fixed-point gap. For the natural VI counterpart, the approximation would be $O_\epsilon(\sqrt{\epsilon})$, and even $O_\epsilon(\epsilon)$ in terms of the *Minty* VI solution (Proposition C.5).

Theorem 3.5 complements the result of Diakonikolas (2025): even though we need $\rho \leq 1 + O_\epsilon(\epsilon^4)$ to get an $\epsilon$-performatively stable point, Theorem 3.5 scales logarithmically with $1/\epsilon$, which is an exponential improvement. On the other hand, the result of Diakonikolas (2025) is not confined to finite-dimensional Euclidean spaces. For an illustration of the different regions in terms of $\rho$ and their complexity, we refer to Figure 1.

Theorem 3.5 is established by showing how to apply the ellipsoid algorithm on a VI problem in which the mapping $F$ is only approximately monotone—specifically, *hypomonotone* (Proposition C.5). Our observation is that, in such problems, *expected* variational inequalities in the sense of Zhang et al. (2025) induce approximate VI solutions, as we formalize in Appendix C.

An interesting open question is whether Theorem 3.5 can be improved to match the approximation-expansiveness tradeoff established by Diakonikolas (2025).

**Unconditional lower bounds** We next establish unconditional query complexity hardness results. In the perfor-

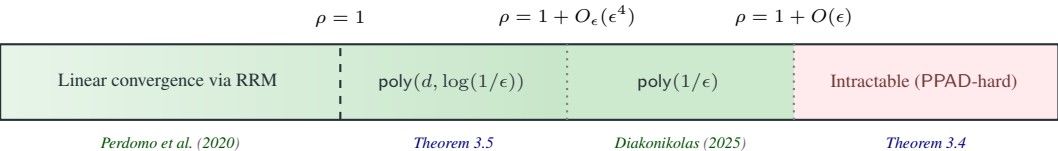

Figure 1. The complexity landscape for computing $\epsilon$-performatively stable points.

mative prediction setting, the natural query model we consider allows an algorithm to specify a point $\boldsymbol{x} \in \mathcal{X}$ and receive the induced RRM iterate $G(\boldsymbol{x})$. We will use the following seminal lower bound due to Hirsch et al. (1989).

**Theorem 3.6** (Hirsch et al., 1989). *For any $d \geq 3$, any algorithm that finds an $\epsilon$-fixed point of a Lipschitz map $T : \mathcal{X} \to \mathcal{X}$ requires at least $c((\frac{1}{\epsilon} - 10)L)^{d-2}$ steps, where $c$ is an absolute constant and $L$ is the Lipschitz constant of $T(\boldsymbol{x}) - \boldsymbol{x}$.*

For the class of problems given in (2)-(3), it follows that $G(\boldsymbol{x}) = g(\boldsymbol{x})$. As a result, under the reduction of Proposition 3.2, every ERM query outputs $(1 - \lambda)\boldsymbol{x} + \lambda T(\boldsymbol{x})$, which reveals as much information as $T(\boldsymbol{x})$ itself.

**Corollary 3.7.** *Computing an $\epsilon$-fixed point of the RRM map $G$ (Definition 1.1) even when $\rho = L\beta/\alpha \leq 1 + O_\epsilon(\epsilon)$ requires $2^{\Omega(d)}$ ERM queries. This holds even when $\epsilon$ is a constant.*

### 3.2. General norms

Having characterized the complexity spectrum in the Euclidean setting (Assumption 2.3), we turn to the more general setting. We extend Assumption 2.3 to general norms (Assumption B.1), and show that the contraction analysis of Perdomo et al. (2020) carries over in this setting.[1]

**Proposition 3.8.** *If $L\beta/\alpha < 1$ per Assumption B.1, the RRM map $G$ (Definition 1.1) is a contraction with respect to $\| \cdot \|$. In particular, if $\boldsymbol{x}^*$ is the unique fixed point,*

$$\|\boldsymbol{x}_t - \boldsymbol{x}^*\| \leq \frac{L\beta}{\alpha}\|\boldsymbol{x}_{t-1} - \boldsymbol{x}^*\| \leq \left(\frac{L\beta}{\alpha}\right)^t \|\boldsymbol{x}_0 - \boldsymbol{x}^*\|.$$

The question now is to characterize the complexity of performatively stable points in this more general setting. The obvious algorithm that arises from Proposition 3.8 computes an $\epsilon$-performatively stable point in a number of iterations that grows as $\log(1/\epsilon)(1 - L\beta/\alpha)^{-1}$. When $1 - L\beta/\alpha \approx 0$, this can be prohibitive.

We first note that, in the regime where $\epsilon$ is not too small, this can be improved using *Halpern iteration* (Halpern, 1967; Lieder, 2021; Wittmann, 1992; Diakonikolas, 2020).

---

[1]With an abuse of notation we use the same symbols to denote the analogous parameters even though convexity, smoothness, and sensitivity are now measured differently. The underlying choice of norm will be clear from the context.

**Corollary 3.9.** *If $L\beta/\alpha \leq 1$ per Assumption B.1, there is an algorithm that finds an $\epsilon$-performatively stable point and has complexity linear in $1/\epsilon$.*

This is another setting in which RRM is inferior to alternative algorithms. The question that remains concerns the complexity when $\epsilon$ is exponentially small. We observe that this is as hard as a major open problem in complexity theory (Condon, 1992; Etessami et al., 2020).

**Proposition 3.10.** *Computing an $\epsilon$-performatively stable point even when the RRM map $G$ (Definition 1.1) satisfies $\|G(\boldsymbol{x}) - G(\boldsymbol{x}')\| < \|\boldsymbol{x} - \boldsymbol{x}'\|$ is as hard as solving a simple stochastic game (SSG).*

For this hardness result, it suffices to consider the $\ell_\infty$ norm, and follows from the fact that the SSG problem reduces to finding fixed points of contractions in the $\ell_\infty$ norm. Unlike our previous results, the precondition of Proposition 3.10 is in terms of the Lipschitz constant of $G$, and not the upper bound $L\beta/\alpha$ (per Assumption B.1).

### 3.3. Complexity for general convex domains

While Theorem 3.4 characterizes the complexity of computing performatively stable points, it relies on prior results established with respect to a hypercube constraint set (that is, $[0, 1]^d$). However, this particular constraint set may not always be aligned with practical applications. For example, in modern machine learning it is often more common to impose an upper bound on the $\ell_2$ norm of the parameter $\boldsymbol{x}$ (Goodfellow et al., 2016; Hinton et al., 2012), which translates to an $\ell_2$ ball constraint set.

To close this gap in the literature, we extend our hardness result to general convex sets under the mild assumption that the domain is *well bounded*; this is a standard regularity condition in convex optimization (Grötschel et al., 1993).

**Definition 3.11** (Well-bounded domains). *A convex and compact set $\mathcal{X}$ is called* well bounded *if there exist $R_1 > 0$ and $R_2 > 0$ such that $\mathcal{B}_{R_1}(\boldsymbol{0}) \subseteq \mathcal{X} \subseteq \mathcal{B}_{R_2}(\boldsymbol{0})$, where $\mathcal{B}_R(\boldsymbol{0})$ is the Euclidean ball centered at $\boldsymbol{0}$ with radius $R$.*

The well-bounded assumption is general and captures many common constraint sets of interest, including the hypercube and the $\ell_2$ ball. Our technical approach only requires that $\mathcal{X} \supseteq \mathcal{B}_{R_1}$ with respect to a two-dimensional ball, so our analysis can encompass sets that may not be

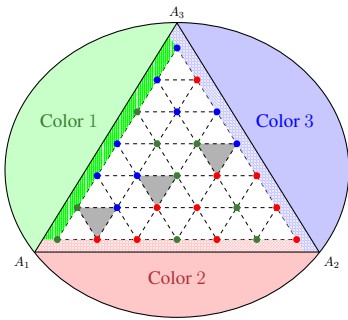
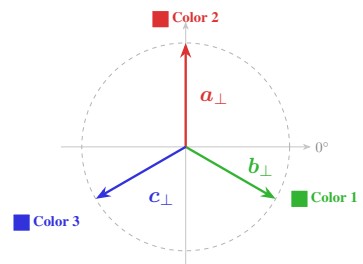

*Figure 2.* Left: Illustration of the proof of Theorem 3.12. Regions outside $\triangle A_1 A_2 A_3$ are mapped with corresponding colors. The shaded bands represent the $\epsilon$-thickness strips used in the construction. Inside $\triangle A_1 A_2 A_3$ the domain forms a triangular grid with colors assigned to the vertices. Trichromatic triangles correspond to VI solutions, and are highlighted by shading. Right: Directional vector mapping induced by the coloring. The directions are chosen so that any point outside $\triangle A_1 A_2 A_3$ is pushed toward the interior of the triangle. As a result, all VI solutions must lie inside $\triangle A_1 A_2 A_3$.

fully dimensional, as is common in machine learning. Furthermore, without loss of generality, the center of the balls can be set at the origin by shifting the domain $\mathcal{X}$.

We establish a complexity hardness result for solving variational inequalities (VIs) over well-bounded domains.

**Theorem 3.12.** *Given a convex and compact domain $\mathcal{X} \subset \mathbb{R}^d$ that is well bounded, an $L$-Lipschitz function $F : \mathcal{X} \to \mathbb{R}^d$, and $\epsilon = O(2^{-n})$, it is PPAD-hard to find a point $\boldsymbol{x}^* \in \mathcal{X}$ such that*

$$\langle \boldsymbol{x} - \boldsymbol{x}^*, F(\boldsymbol{x}^*) \rangle \leq \epsilon \quad \forall \boldsymbol{x} \in \mathcal{X}. \tag{4}$$

*This holds even when $d = 2$ and $L = O(1)$.*

This result is of broader interest in the complexity of variational inequalities. The requirement that $\epsilon$ is exponentially small is necessary to prove hardness in low dimensions. In contrast, high-precision solutions can be attained in the contractive regime since RRM exhibits linear convergence.

Theorem 3.12 follows from constructing a polynomial-time reduction from the 2D-SPERNER problem, which was shown to be PPAD-complete by Chen & Deng (2009). Our proof proceeds by showing that, given a well-bounded domain $\mathcal{X}$, one can locate an equilateral triangle $\triangle A_1 A_2 A_3 \subseteq \mathcal{X}$, which will serve as the domain for the 2D-SPERNER instance. To construct a continuous mapping $F : \mathcal{X} \to \mathcal{X}$ from the coloring of the 2D-SPERNER problem, we carefully design an arithmetic circuit that converts the coloring of a given point to vectors in Euclidean space. A key technical challenge is that the arithmetic circuit $F$ is continuous, whereas the coloring in the 2D-SPERNER problem is specified by a boolean circuit and is therefore discontinuous. As a result, $F$ cannot exactly match the coloring everywhere in $\mathcal{X}$. We address this issue by employing a sampling technique introduced by Deligkas et al. (2020) which allows us to control the error arising from this mismatch. Figure 2 illustrates the coloring and the choice of

directional vectors respectively. We defer the detailed proof to Appendix D.

We next apply our general result to the problem of computing performatively stable points. We show that, even under a general constraint set, PPAD-hardness persists for some accuracy $\epsilon = O(2^{-n})$ when $\rho > 1$. We summarize this result below.

**Corollary 3.13.** *For any convex compact domain $\mathcal{X}$ that is well-bounded, finding an $\epsilon$-performatively stable point per Definition 2.2 is PPAD-hard even when $L\beta/\alpha \leq 1 + \frac{\epsilon}{\epsilon'}$ for some $\epsilon' = O(2^{-n})$. This is so even when $\ell$ is a quadratic objective, $\ell(\boldsymbol{x}; \boldsymbol{z}) = \frac{1}{2}\|\boldsymbol{x} - \boldsymbol{z}\|_2^2$.*

The proof is similar to Theorem 3.4, using $g(\boldsymbol{x}) = \boldsymbol{x} + \frac{\epsilon}{\epsilon'} F(\boldsymbol{x})$, where $F(\boldsymbol{x})$ is the operator in Theorem 3.12.

*Remark* 3.14. By choosing $\epsilon < \epsilon' = O(2^{-n})$, the expansion parameter $L\beta/\alpha$ can be made arbitrarily close to $1$, while PPAD-hardness still persists.

## 4. Strategic classification

As highlighted in Section 1.2, performative prediction encompasses the problem of strategic classification. The complexity of computing performatively optimal points—also known as "strategic maxima" in this line of work—was already shown to be NP-hard in the original paper by Hardt et al. (2016). Two natural questions arise: i) what is the complexity of *local* performative optimality? And ii) what is the complexity of performative stability in strategic classification? Concerning the second question, the class of problems we have considered so far is no longer suitable since strategic classification is a more structured problem. Let us first recall the definition of strategic classification.

**Definition 4.1** (Strategic classification; Hardt et al., 2016)**.** Strategic classification is a game played between the *Jury* and the *Contestant*. Let $\mathcal{D}$ be a distribution over a population $X$, $c : X \times X \to \mathbb{R}_{\geq 0}$ a cost function, and $h$ a target

classifier.

1. The Jury first publishes a classifier $f : X \to \{0, 1\}$, which may depend on the cost function $c$, the distribution $\mathcal{D}$, and the target classifier $h$.

2. The Contestant, who knows $c$, $h$, $\mathcal{D}$, and $f$, selects a deviation $\Delta : X \to X$.

The payoff to the Jury is $\Pr_{\boldsymbol{x} \sim \mathcal{D}}[h(\boldsymbol{x}) = f(\Delta(\boldsymbol{x}))]$ and the payoff to the Contestant is $\mathbb{E}_{\boldsymbol{x} \sim \mathcal{D}}[f(\Delta(\boldsymbol{x})) - c(\boldsymbol{x}, \Delta(\boldsymbol{x}))]$.

Strategic classification is commonly formulated as a Stackelberg game between the Jury and the Contestant, in which the Contestant always best-responds to the classifier published by the Jury, while the Jury seeks to maximize their utility while accounting for the strategic deviation of the Contestant. The strategic maximum is defined as follows.

**Definition 4.2** (Strategic maximum; Hardt et al., 2016). Given a population $X$, a distribution $\mathcal{D}$ over $X$, a cost function $c : X \times X \to \mathbb{R}_{\geq 0}$, and a target classifier $h$, a classifier $f^*$ for the Jury is said to be at *strategic maximum* if

$$f^* \in \operatorname*{argmax}_{f:X \to \{0,1\}} \Pr_{\boldsymbol{x} \sim \mathcal{D}}[h(\boldsymbol{x}) = f(\Delta(\boldsymbol{x}))].$$

**Strategic local optimality**  Finding a global optimum in strategic classification has a wide range of applications, but is computationally intractable. In particular, Hardt et al. (2016) established NP-completeness. A natural question is to consider local search algorithms for the Jury, who can explore local moves into nearby configurations with the goal to converge to a local optimum.

We begin by formally defining the notion of local moves for the Jury. Consider a finite population $X$ with $|X| = n$, and a classifier $f$. We say that the Jury makes a local move to a nearby configuration by updating the label of a single data point. Formally, a classifier $f'$ is a nearby configuration of $f$ if there exists an index $i \in [n]$ such that

$$f'(X) = f(X) \oplus \boldsymbol{e}_i,$$

where $\oplus$ denotes the XOR operation and $\boldsymbol{e}_i$ is the $i$th standard basis vector. We denote by $\mathcal{N}(f)$ the set of all classifiers that can be reached from $f$ with one local move. We now define the notion of strategic local optimum.

**Definition 4.3.** Given a finite population $X$, a distribution $\mathcal{D}$ over $X$, a cost function $c$, and a target classifier $h$, a classifier $f^*$ is said to be at *strategic local optimum* if

$$\Pr_{\boldsymbol{x} \sim \mathcal{D}}[h(\boldsymbol{x}) = f^*(\Delta(\boldsymbol{x}))] \geq \max_{f \in \mathcal{N}(f^*)} \Pr_{\boldsymbol{x} \sim \mathcal{D}}[h(\boldsymbol{x}) = f(\Delta(\boldsymbol{x}))].$$

We proceed to state the main result of this section.

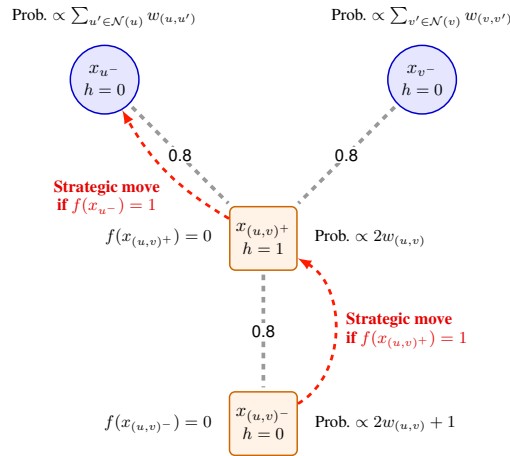

*Figure 3.* Our basic edge gadget for edge $(u, v)$. The Jury would like to classify $x_{(u,v)^+}$ as 1, but $x_{(u,v)^-}$ would then strategically deviate to $x_{(u,v)^+}$. This forces the Jury to pick a classifier such that $f(x_{(u,v)^+}) = 0 = f(x_{(u,v)^-})$. Furthermore, if the Jury switches the label of $x_{u^-}$ from 0 to 1, all edges incident to $u$ in the graph that were previously classified as 0 can profitably deviate to $x_{u^-}$. The change in the Jury's utility reflects the change in the weight of the cut induced by $f$.

**Theorem 4.4.** *Given a finite population $X$, a distribution $\mathcal{D}$ over $X$, a cost function $c$, and a target classifier $h$, it is PLS-hard to find a strategic local optimum as in Definition 4.3. This result holds even when $c$ is a metric and the target classifier $h$ is provided explicitly to the algorithm.*

Theorem 4.4 shows that, unless there is a collapse in the complexity hierarchy (specifically, P = PLS), finding even a local optimum in strategic classification cannot be achieved in polynomial time.

The proof of Theorem 4.4 takes an instance of LO­CALMAXCUT problem, which was shown to be PLS­complete by Schäffer & Yannakakis (1991), and constructs a polynomial-time reduction to the problem of finding a strategic local optimum.

In particular, let $G = (V, E, w)$ be a weighted undirected graph with edge weights $w_{(u,v)} \geq 0$ for any edge $(u, v) \in E$. We construct an instance of strategic classification consisting of a finite population $X$ and a non-uniform distribution $\mathcal{D}$ over $X$. For each vertex $v \in V$, we introduce a point $x_{v^-}$ with label $h(x_{v^-}) = 0$. For each edge $(u, v) \in E$, we introduce two points $x_{(u,v)^+}$ and $x_{(u,v)^-}$ with labels $h(x_{(u,v)^+}) = 1$ and $h(x_{(u,v)^-}) = 0$. We design the cost function $c$ with specific values such that the admissible deviations of the Contestant satisfy the following: the point $x_{(u,v)^-}$ may deviate only to $x_{(u,v)^+}$, and the vertex points $x_{u^-}$ and $x_{v^-}$ may deviate only to $x_{(u,v)^+}$. Figure 3 illustrates the gadget for a simple graph with two vertices and one edge, where the dashed edges represent admissible deviations for the Contestant.

The first observation is that for any classifier $f^*$ that is at a strategic optimum, $f^*(x_{(u,v)^-}) = 0$ and $f^*(x_{(u,v)^+}) = 0$. This follows from the fact that the distribution $\mathcal{D}$ assigns higher probability to $x_{(u,v)^-}$ than to $x_{(u,v)^+}$. As a result, any classifier that labels either $x_{(u,v)^+}$ or $x_{(u,v)^-}$ as positive will incur a net loss due to misclassifying $x_{(u,v)^-}$.

The result is that the Jury will only label vertex points $x_{v^-}$ as positive and improve their utility through the deviation of $x_{(u,v)^+}$ to the corresponding vertex points $x_{u^-}$ or $x_{v^-}$. Through a careful design of the distribution weights, we show that any strategically local optimal classifier $f^*$ induces a cut of the original graph $G$: vertices $v \in V$ with $f^*(x_{v^-}) = 0$ lie on one side of the cut, while vertices $u \in V$ with $f^*(x_{u^-}) = 1$ lie on the other. Moreover, the strategic local optimality of $f^*$ ensures the local optimality of the induced cut. We defer the full proof to Appendix E.

Notably, our construction in Theorem 4.4 based on MAX-CUT also yields an alternative proof that finding a global strategic optimum is NP-complete (Hardt et al., 2016).

**Performative stability with endogenous costs** Moreover, we establish hardness for computing a Nash equilibrium—which translates to a performatively stable point—in an extension of Definition 4.1 in which the Jury can also affect the cost incurred by deviating. This extension captures the fact that, in reality, decision-makers often act as regulators who go beyond merely classifying (Alhanouti et al., 2025); they actively dictate the cost structure by imposing sanctions and penalizing more certain deviation types, for example through audits (Estornell et al., 2021). We refer to this setting as strategic classification with *endogenous costs* (Definition F.3). Compared to Definition 4.1, our definition also posits that the Jury is constrained to select a classifier from a specified set of classifiers, and similarly for the Contestant. We find that this class of problems is rich enough to encode PPAD-hard problems.

**Proposition 4.5.** *Computing a performatively stable point in strategic classification with endogenous costs is* PPAD-*hard.*

The proof is based on a reduction from a win-loss two-player general-sum game $(\mathbf{A}, \mathbf{B})$ (Abbott et al., 2005). We associate each column to a separate classifier and each row to a point drawn uniformly from $X$. Each classifier $f_j$ labels the points in accordance with $\mathbf{B}_{:,j}$. The target classifier is taken to be $h(\boldsymbol{x}) = 0$ for all $\boldsymbol{x} \in X$. The Contestant can pick any constant deviation, and the payoff matrix $\mathbf{A}$ is encoded through the costs, making them the dominant component of the Contestant's utility. The detailed argument is deferred to Appendix F.

This reduction heavily relies on the presence of endogenous costs. The complexity of computing performatively

stable points based on the more common Definition 4.1 remains an open problem. Because of the particular payoff structure, we suspect that the latter problem may be easier.

*Remark* 4.6. While our hardness results are worst case in nature, they have clear algorithmic implications: to bypass those complexity barriers, one should either restrict to more structured classes of instances or examine tractable relaxations of performatively stable points; this latter direction was pursued by Farina & Perdomo (2026) subsequently to our paper.

## 5. Conclusions and future research

We have established a sharp computational phase transition for performative stability, showing that while slight expansiveness is tractable, the problem quickly becomes PPAD-hard. We also characterized the complexity of computing local strategic maxima in strategic classification. An important question that arises from our results is to characterize the complexity of computing performatively stable points in strategic classification per Definition 4.1. Another interesting avenue for future research is to close the gap between the expansiveness tolerance of our ellipsoid-based approach (Theorem 3.5), the recent result of Diakonikolas (2025), and the PPAD-hardness established in Theorem 3.4, thereby refining the complexity landscape that emerged from our paper (Figure 1).

## Acknowledgments

Ioannis Panageas is supported by NSF grant CCF-2454115. Tuomas Sandholm is supported by NIH award A240108S001, the Vannevar Bush Faculty Fellowship ONR N00014-23-1-2876, and National Science Foundation grant RI-2312342.

## Impact statement

This paper presents work whose goal is to advance the theoretical foundations of machine learning. We do not foresee any immediate societal consequences or ethical concerns.

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

# A. First-order performative stability

In this section, we examine the notion of first-order approximate performative stability, which will be relevant for analyzing the computational complexity of finding performatively stable points. We note that computing a performatively stable point can be viewed as a fixed point computation problem (Lemma F.2). Under mild assumptions on the induced distribution $\mathcal{D}(\cdot)$, if the objective function $\ell(\boldsymbol{x}; \boldsymbol{z})$ is convex in $\boldsymbol{x}$ for all $\boldsymbol{z}$ and jointly continuous in $(\boldsymbol{x}, \boldsymbol{z})$, a performatively stable point is guaranteed to exist (Perdomo et al., 2020). However, when the objective function $\ell(\boldsymbol{x}; \boldsymbol{z})$ is nonconvex in $\boldsymbol{x}$, the argmin operator in Definition 2.2 may not be efficiently computable. To address this issue, we consider an alternative definition, which we restate below.

**Definition A.1** (Performative stability). A point $\boldsymbol{x}^* \in \mathcal{X}$ is (first-order) performatively stable if for all $\boldsymbol{x} \in \mathcal{X}$, it holds that

$$\left\langle \boldsymbol{x} - \boldsymbol{x}^*, \mathbb{E}_{\boldsymbol{z} \sim \mathcal{D}(\boldsymbol{x}^*)}\left[\nabla_{\boldsymbol{x}} \ell(\boldsymbol{x}^*; \boldsymbol{z})\right]\right\rangle \geq 0.$$

By virtue of existing complexity results pertaining to variational inequalities, computing a first-order performatively stable point lies in PPAD under mild assumptions on the representation of $\mathcal{X}$, $\mathcal{D}$, and $\ell$. This complements Theorem 3.4.

**Corollary A.2.** *Computing an $\epsilon$-performatively stable point is in* PPAD.

As we now show, when $\ell(\boldsymbol{x}; \boldsymbol{z})$ is convex in $\boldsymbol{x}$ for all $\boldsymbol{z}$, the notion of first-order performative stability coincides with the notion of performative stability per Definition 2.2.

**Claim A.3.** *If $\ell(\boldsymbol{x}; \boldsymbol{z})$ is convex in $\boldsymbol{x}$ for any $\boldsymbol{z}$, $\boldsymbol{x}^* \in \mathcal{X}$ is a performatively stable point if and only if $\boldsymbol{x}^*$ is a first-order performatively stable point.*

*Proof.* Let $\boldsymbol{x}^* \in \mathcal{X}$ be a first-order performatively stable point, since $\ell(\boldsymbol{x}; \boldsymbol{z})$ is convex in $\boldsymbol{x}$, for any $\boldsymbol{x}' \in \mathcal{X}$, we have

$$\mathbb{E}_{\boldsymbol{z} \sim \mathcal{D}(\boldsymbol{x}^*)}\left[\ell(\boldsymbol{x}'; \boldsymbol{z})\right] - \mathbb{E}_{\boldsymbol{z} \sim \mathcal{D}(\boldsymbol{x}^*)}\left[\ell(\boldsymbol{x}^*; \boldsymbol{z})\right] \geq \mathbb{E}_{\boldsymbol{z} \sim \mathcal{D}(\boldsymbol{x}^*)}\left[\left\langle \boldsymbol{x}' - \boldsymbol{x}^*, \nabla_{\boldsymbol{x}} \ell(\boldsymbol{x}^*; \boldsymbol{z})\right\rangle\right]$$
$$= \left\langle \boldsymbol{x}' - \boldsymbol{x}^*, \mathbb{E}_{\boldsymbol{z} \sim \mathcal{D}(\boldsymbol{x}^*)}\left[\nabla_{\boldsymbol{x}} \ell(\boldsymbol{x}^*; \boldsymbol{z})\right]\right\rangle \qquad (5)$$
$$\geq 0, \qquad (6)$$

where in (6) we use the definition of first-order performative stability. On the other hand, suppose $\boldsymbol{x}^* \in \mathcal{X}$ is a performatively stable point, for any $\boldsymbol{x}' \in \mathcal{X}$ and any $\alpha \in (0, 1]$, it holds that

$$\mathbb{E}_{\boldsymbol{z} \sim \mathcal{D}(\boldsymbol{x}^*)}\left[\ell(\boldsymbol{x}^*; \boldsymbol{z})\right] \leq \mathbb{E}_{\boldsymbol{z} \sim \mathcal{D}(\boldsymbol{x}^*)}\left[\ell(\boldsymbol{x}^* + \alpha(\boldsymbol{x}' - \boldsymbol{x}^*); \boldsymbol{z})\right].$$

By Taylor's theorem

$$\mathbb{E}_{\boldsymbol{z} \sim \mathcal{D}(\boldsymbol{x}^*)}\left[\ell(\boldsymbol{x}^* + \alpha(\boldsymbol{x}' - \boldsymbol{x}^*); \boldsymbol{z})\right] - \mathbb{E}_{\boldsymbol{z} \sim \mathcal{D}(\boldsymbol{x}^*)}\left[\ell(\boldsymbol{x}^*; \boldsymbol{z})\right] = \alpha\left\langle \boldsymbol{x}' - \boldsymbol{x}^*, \mathbb{E}_{\boldsymbol{z} \sim \mathcal{D}(\boldsymbol{x}^*)}\left[\nabla_{\boldsymbol{x}} \ell(\boldsymbol{x}^*; \boldsymbol{z})\right]\right\rangle + o(\alpha).$$

If $\left\langle \boldsymbol{x}' - \boldsymbol{x}, \mathbb{E}_{\boldsymbol{z} \sim \mathcal{D}(\boldsymbol{x}^*)}\left[\nabla_{\boldsymbol{x}} \ell(\boldsymbol{x}^*; \boldsymbol{z})\right]\right\rangle < 0$, it would imply that for a sufficiently small $\alpha > 0$, we have $\mathbb{E}_{\boldsymbol{z} \sim \mathcal{D}(\boldsymbol{x}^*)}\left[\ell(\boldsymbol{x}^* + \alpha(\boldsymbol{x}' - \boldsymbol{x}^*); \boldsymbol{z})\right] - \mathbb{E}_{\boldsymbol{z} \sim \mathcal{D}(\boldsymbol{x}^*)}\left[\ell(\boldsymbol{x}^*; \boldsymbol{z})\right] < 0$, contradicting the fact that $\boldsymbol{x}^*$ is a performatively stable point per Definition 2.2. The proof is complete. $\qquad \square$

# B. Contraction for general norms

This section generalizes the contraction proof of Perdomo et al. (2020) from the $\|\cdot\|_2$ norm to arbitrary norms. In particular, we adapt Assumption 2.3 as follows.

**Assumption B.1.** *Let $\ell(\boldsymbol{x}; \boldsymbol{z})$ be the loss function and $\mathcal{D}(\boldsymbol{x})$ the distribution on $\mathcal{Z}$ induced by $\boldsymbol{x} \in \mathcal{X}$.*

- *(strong convexity) $\ell(\boldsymbol{x}; \boldsymbol{z})$ is $\alpha$-strongly convex with respect to $\|\cdot\|$:*

$$\ell(\boldsymbol{x}; \boldsymbol{z}) \geq \ell(\boldsymbol{x}'; \boldsymbol{z}) + \left\langle \nabla_{\boldsymbol{x}} \ell(\boldsymbol{x}'; \boldsymbol{z}), \boldsymbol{x} - \boldsymbol{x}'\right\rangle + \frac{\alpha}{2}\|\boldsymbol{x} - \boldsymbol{x}'\|^2$$

*for any $\boldsymbol{x}, \boldsymbol{x}' \in \mathcal{X}$ and $\boldsymbol{z} \in \mathcal{Z}$.*

- *(smoothness) $\ell(\boldsymbol{x}; \boldsymbol{z})$ is $\beta$(-jointly) smooth if*

$$\|\nabla_{\boldsymbol{x}}\ell(\boldsymbol{x}; \boldsymbol{z}) - \nabla_{\boldsymbol{x}}\ell(\boldsymbol{x}'; \boldsymbol{z})\|_* \leq \beta\|\boldsymbol{x} - \boldsymbol{x}'\|$$

  *and*

$$\|\nabla_{\boldsymbol{x}}\ell(\boldsymbol{x}; \boldsymbol{z}) - \nabla_{\boldsymbol{x}}\ell(\boldsymbol{x}; \boldsymbol{z}')\|_* \leq \beta\|\boldsymbol{z} - \boldsymbol{z}'\|$$

  *for any $\boldsymbol{x}, \boldsymbol{x}' \in \mathcal{X}$ and $\boldsymbol{z}, \boldsymbol{z}' \in \mathcal{Z}$.*

- *(sensitivity) $\mathcal{D}$ is $L$-sensitive if*

$$W_1(\mathcal{D}(\boldsymbol{x}), \mathcal{D}(\boldsymbol{x}')) \leq L\|\boldsymbol{x} - \boldsymbol{x}'\|$$

  *for any $\boldsymbol{x}, \boldsymbol{x}' \in \mathcal{X}$, where $W_1$ denotes the Wasserstein-1 distance, or earth mover's distance.*

Above, we denote by $\|\cdot\|_*$ the dual norm of $\|\cdot\|$. We point out that the contraction argument of Perdomo et al. (2020) readily carries over under Assumption B.1.

**Proposition 3.8.** *If $L\beta/\alpha < 1$ per Assumption B.1, the RRM map $G$ (Definition 1.1) is a contraction with respect to $\|\cdot\|$. In particular, if $\boldsymbol{x}^*$ is the unique fixed point,*

$$\|\boldsymbol{x}_t - \boldsymbol{x}^*\| \leq \frac{L\beta}{\alpha}\|\boldsymbol{x}_{t-1} - \boldsymbol{x}^*\| \leq \left(\frac{L\beta}{\alpha}\right)^t \|\boldsymbol{x}_0 - \boldsymbol{x}^*\|.$$

*Proof.* Let $\boldsymbol{x}, \boldsymbol{x}' \in \mathcal{X}$, $f(\boldsymbol{y}) = \mathbb{E}_{\boldsymbol{z} \sim \mathcal{D}(\boldsymbol{x})}\ell(\boldsymbol{y}; \boldsymbol{z})$, and $f'(\boldsymbol{y}) = \mathbb{E}_{\boldsymbol{z} \sim \mathcal{D}(\boldsymbol{x}')}\ell(\boldsymbol{y}; \boldsymbol{z})$. Taking the expectation over $\boldsymbol{z} \sim \mathcal{D}(\boldsymbol{x})$, it follows that $f(\boldsymbol{y})$ is $\alpha$-strongly convex with respect to $\|\cdot\|$. Thus,

$$f(G(\boldsymbol{x})) \geq f(G(\boldsymbol{x}')) + \langle G(\boldsymbol{x}) - G(\boldsymbol{x}'), \nabla f(G(\boldsymbol{x}'))\rangle + \frac{\alpha}{2}\|G(\boldsymbol{x}) - G(\boldsymbol{x}')\|^2, \tag{7}$$

where $G$ is the RRM mapping (Definition 1.1). Since $G(\boldsymbol{x})$ is, by definition, the unique minimizer of $f$, we also have $\langle G(\boldsymbol{x}') - G(\boldsymbol{x}), \nabla f(G(\boldsymbol{x}))\rangle \geq 0$ by the first-order optimality condition. In turn, this implies

$$f(G(\boldsymbol{x}')) \geq f(G(\boldsymbol{x})) + \frac{\alpha}{2}\|G(\boldsymbol{x}) - G(\boldsymbol{x}')\|^2. \tag{8}$$

Combining (7) and (8), we have

$$\langle G(\boldsymbol{x}') - G(\boldsymbol{x}), \nabla f(G(\boldsymbol{x}'))\rangle \geq \alpha\|G(\boldsymbol{x}) - G(\boldsymbol{x}')\|^2. \tag{9}$$

Furthermore, $\langle G(\boldsymbol{x}') - G(\boldsymbol{x}), \nabla\ell(G(\boldsymbol{x}'); \boldsymbol{z})\rangle$ is $(\|G(\boldsymbol{x}') - G(\boldsymbol{x})\|\beta)$-Lipschitz continuous in $\boldsymbol{z}$ since

$$|\langle G(\boldsymbol{x}') - G(\boldsymbol{x}), \nabla\ell(G(\boldsymbol{x}'); \boldsymbol{z})\rangle - \langle G(\boldsymbol{x}') - G(\boldsymbol{x}), \nabla\ell(G(\boldsymbol{x}'); \boldsymbol{z}')\rangle|$$
$$\leq \|G(\boldsymbol{x}') - G(\boldsymbol{x})\|\|\nabla\ell(G(\boldsymbol{x}'); \boldsymbol{z}) - \nabla\ell(G(\boldsymbol{x}'); \boldsymbol{z}')\|_*$$
$$\leq \beta\|G(\boldsymbol{x}') - G(\boldsymbol{x})\|,$$

by $\beta$-joint smoothness. Now, for the distribution map $\mathcal{D}(\cdot)$, Kantorovich-Rubinstein duality yields

$$\left|\mathbb{E}_{\boldsymbol{z} \sim \mathcal{D}(\boldsymbol{x})}g(\boldsymbol{z}) - \mathbb{E}_{Z \sim \mathcal{D}(\boldsymbol{x}')}g(\boldsymbol{z})\right| \leq L\|\boldsymbol{x} - \boldsymbol{x}'\| \quad \forall g \text{ 1-Lipschitz.}$$

As a result,

$$\langle G(\boldsymbol{x}) - G(\boldsymbol{x}'), \nabla f(G(\boldsymbol{x}'))\rangle - \langle G(\boldsymbol{x}) - G(\boldsymbol{x}'), \nabla f'(G(\boldsymbol{x}'))\rangle \geq -L\beta\|G(\boldsymbol{x}') - G(\boldsymbol{x})\|\|\boldsymbol{x} - \boldsymbol{x}'\|.$$

By the first-order optimality condition, it also follows that $\langle G(\boldsymbol{x}) - G(\boldsymbol{x}'), \nabla f'(G(\boldsymbol{x}'))\rangle \geq 0$ since $G(\boldsymbol{x}')$ is the minimizer of $f'$. So,

$$\langle G(\boldsymbol{x}) - G(\boldsymbol{x}'), \nabla f(G(\boldsymbol{x}'))\rangle \geq -L\beta\|G(\boldsymbol{x}') - G(\boldsymbol{x})\|\|\boldsymbol{x} - \boldsymbol{x}'\|. \tag{10}$$

Combining (9) and (10), we conclude that

$$L\beta\|G(\boldsymbol{x}') - G(\boldsymbol{x})\|\|\boldsymbol{x} - \boldsymbol{x}'\| \geq \alpha\|G(\boldsymbol{x}) - G(\boldsymbol{x}')\|^2 \implies \|G(\boldsymbol{x}) - G(\boldsymbol{x}')\| \leq \frac{L\beta}{\alpha}\|\boldsymbol{x} - \boldsymbol{x}'\|.$$

In other words, if $L\beta/\alpha < 1$, $G$ is a contraction with respect to the norm $\|\cdot\|$, as claimed. $\square$

Similar extensions are possible for other algorithms beyond repeated risk minimization, such as repeated gradient descent.

## C. Ellipsoid for Euclidean expansive mappings

A well-known result in optimization is that there is a polynomial-time algorithm for computing fixed points of nonexpansive mappings *with respect to the $\ell_2$ norm* (Huang et al., 1999; Sikorski et al., 1993); the complexity of this problem is a major open question for more general norms. In particular, for mappings that are contracting with respect to the $\ell_2$ norm, there is an algorithm whose complexity does not depend on the contraction parameter. In the setting of performative prediction, we begin by observing that this can be used to obtain a significant improvement in the setting where $\beta L/\alpha \approx 1$. The number of iterations needed to reach an approximate fixed point under repeated risk minimization is proportional to $\log(1/\epsilon)\frac{1}{\frac{\alpha}{\beta L}-1}$ in the regime where $\beta L/\alpha \approx 1$, thereby blowing up.

**Theorem C.1** (Sikorski et al., 1993). *Consider a continuous mapping $T : \mathcal{X} \to \mathcal{X}$, where $\mathcal{X}$ is a subset of the $d$-dimensional Euclidean space, that is nonexpansive with respect to the $\ell_2$ norm; that is, $\|T(\boldsymbol{x}) - T(\boldsymbol{x}')\|_2 \le \|\boldsymbol{x} - \boldsymbol{x}'\|_2$ for any $\boldsymbol{x}, \boldsymbol{x}' \in \mathcal{X}$. Then there is a $\mathsf{poly}(d, \log(1/\epsilon))$-time algorithm that computes an $\epsilon$-fixed point of $T$.*

We provide the simple proof below, as we will use a similar bound in our extension.

*Proof of Theorem C.1.* We will prove that the operator $F : \boldsymbol{x} - T(\boldsymbol{x})$ is monotone. That is, $\langle F(\boldsymbol{x}) - F(\boldsymbol{x}'), \boldsymbol{x} - \boldsymbol{x}' \rangle \ge 0$ for any $\boldsymbol{x}, \boldsymbol{x}' \in \mathcal{X}$. Indeed, we write

$$\langle F(\boldsymbol{x}) - F(\boldsymbol{x}'), \boldsymbol{x} - \boldsymbol{x}' \rangle = \frac{1}{2}\left(\|\boldsymbol{x} - \boldsymbol{x}'\|_2^2 - \|T(\boldsymbol{x}) - T(\boldsymbol{x}')\|_2^2 + \|\boldsymbol{x} - T(\boldsymbol{x}) - \boldsymbol{x}' + T(\boldsymbol{x}')\|_2^2\right) \ge 0 \qquad (11)$$

since $\|\boldsymbol{x} - \boldsymbol{x}'\|_2 \ge \|T(\boldsymbol{x}) - T(\boldsymbol{x}')\|_2$ and $\|\cdot\|_2 \ge 0$. Now, let $\boldsymbol{x}$ be an $\epsilon$-approximate VI solution with respect to $F$, which means that $\langle \boldsymbol{x}' - \boldsymbol{x}, F(\boldsymbol{x}) \rangle \ge -\epsilon$ for any $\boldsymbol{x}' \in \mathcal{X}$. In particular, setting $\boldsymbol{x}' = T(\boldsymbol{x})$ yields $\langle T(\boldsymbol{x}) - \boldsymbol{x}, \boldsymbol{x} - T(\boldsymbol{x}) \rangle \ge -\epsilon$, or $-\|\boldsymbol{x} - T(\boldsymbol{x})\|_2^2 \ge -\epsilon$, which is to say that $\boldsymbol{x}$ is a $\sqrt{\epsilon}$-fixed point of $T$. Moreover, an $\epsilon$-approximate VI solution with respect to $F$ can be computed in time $\mathsf{poly}(d, \log(1/\epsilon))$ since $F$ is monotone. This completes the proof. $\qquad\square$

There is also a more direct argument that does not go through the monotonicity of the gap function. In particular, it is possible to develop a separation oracle by relying on the fact that $T$ is nonexpansive: for any point $\boldsymbol{x}_k \in \mathcal{X}$, first test whether $T(\boldsymbol{x}_k) = \boldsymbol{x}_k$. If not, the key observation is that $\boldsymbol{g}_k = \boldsymbol{x}_k - T(\boldsymbol{x}_k)$ serves as a separating hyperplane. This is so because $\langle \boldsymbol{x}_k - \boldsymbol{x}, \boldsymbol{g}_k \rangle \ge 0$ for any $\boldsymbol{x} \in \mathcal{X}$ that is a fixed point of $T$; since $\langle \boldsymbol{x}_k - \boldsymbol{x}, \boldsymbol{g}_k \rangle = \langle \boldsymbol{x}_k - \boldsymbol{x}, F(\boldsymbol{x}_k) \rangle = \langle \boldsymbol{x}_k - \boldsymbol{x}, F(\boldsymbol{x}_k) - F(\boldsymbol{x}) \rangle \ge 0$ by (11).

**Corollary C.2.** *If $\rho = L\beta/\alpha \le 1$ (per Assumption 2.3), there is a $\mathsf{poly}(d, \log(1/\epsilon))$-time algorithm for computing an $\epsilon$-performatively stable point.*

To put this into better context, it is important to point out that repeated risk minimization can fail when $\rho = 1$. For completeness, we include the simple example below.

*Example* C.3 (Cycling dynamics at the threshold). Consider a one-dimensional setting where $\mathcal{X}$ is centrally symmetric and the loss is the simple quadratic objective $\ell(\boldsymbol{x}; \boldsymbol{z}) = \frac{1}{2}\|\boldsymbol{x} - \boldsymbol{z}\|^2$. This function is 1-jointly smooth and 1-strongly convex ($\beta = \alpha = 1$). Suppose further that the distribution $\mathcal{D}(\boldsymbol{x})$ is a point mass supported on $\boldsymbol{z} = g(\boldsymbol{x}) \coloneqq -\boldsymbol{x}$. The sensitivity of this map is $L = 1$, resulting in $\rho = L\beta/\alpha = 1$.

The repeated risk minimization (RRM) update at step $t$ minimizes the loss on the distribution induced by the current iterate $\boldsymbol{x}_t$. Since the distribution is supported on $\boldsymbol{z} = -\boldsymbol{x}_t$, the update becomes:

$$\boldsymbol{x}_{t+1} = \underset{\boldsymbol{x} \in \mathcal{X}}{\arg\min} \frac{1}{2}\|\boldsymbol{x} - (-\boldsymbol{x}_t)\|^2 = -\boldsymbol{x}_t. \qquad (12)$$

Starting from any initialization $\boldsymbol{x}_0 \ne 0$, the algorithm oscillates indefinitely between $\boldsymbol{x}_0$ and $-\boldsymbol{x}_0$, failing to converge to the unique performatively stable point $\boldsymbol{x}^* = 0$.

**Extension to expansive mappings** Interestingly, we observe that Theorem C.1 can be extended when $T$ can be marginally expansive. Let us first present an approach that works for monotone operators, and we shall then relax the monotonicity assumption. We rely on the notion of an *expected variational inequality (EVI)* (Zhang et al., 2025). In particular, an $\epsilon$-EVI solution $\mu \in \Delta(\mathcal{X})$ satisfies

$$\mathbb{E}_{\boldsymbol{x} \sim \mu}[\langle F(\boldsymbol{x}), \boldsymbol{x} - \boldsymbol{x}' \rangle] \le \epsilon \quad \forall \boldsymbol{x}' \in \mathcal{X}. \qquad (13)$$

Zhang et al. (2025) gave a $\mathsf{poly}(d, \log(1/\epsilon))$ for computing an $\epsilon$-EVI solution. We will first argue that, for monotone operators, the mean of the distribution $\overline{\boldsymbol{x}} = \mathbb{E}_{\boldsymbol{x} \sim \mu}[\boldsymbol{x}]$ is an $\epsilon$-approximate solution to the *Minty* VI problem:

$$\langle F(\boldsymbol{x}'), \boldsymbol{x}' - \overline{\boldsymbol{x}} \rangle \geq -\epsilon \quad \forall \boldsymbol{x}' \in \mathcal{X}. \tag{14}$$

Indeed, starting from (13) and using monotonicity, we have that for any $\boldsymbol{x}' \in \mathcal{X}$,

$$
\begin{aligned}
\epsilon &\geq \mathbb{E}_{\boldsymbol{x} \sim \mu}[\langle F(\boldsymbol{x}), \boldsymbol{x} - \boldsymbol{x}' \rangle] \\
&\geq \mathbb{E}_{\boldsymbol{x} \sim \mu}[\langle F(\boldsymbol{x}'), \boldsymbol{x} - \boldsymbol{x}' \rangle] \\
&= \langle F(\boldsymbol{x}'), \mathbb{E}_{\boldsymbol{x} \sim \mu}[\boldsymbol{x}] - \boldsymbol{x}' \rangle \\
&= \langle F(\boldsymbol{x}'), \overline{\boldsymbol{x}} - \boldsymbol{x}' \rangle.
\end{aligned}
$$

Rearranging, this establishes (14). Finally, to go from an $\epsilon$-MVI solution to an approximate VI solution, we use the following standard lemma.

**Lemma C.4** (Relation between $\epsilon$-MVI and SVI). *Let $F : \mathcal{X} \to \mathbb{R}^d$ be an operator that is L-Lipschitz continuous. If $\boldsymbol{x}^* \in \mathcal{X}$ is an $\epsilon$-approximate MVI solution, then $\boldsymbol{x}^*$ is an $O_\epsilon(\sqrt{\epsilon})$-approximate (Stampacchia) VI solution. Specifically, if $D$ is the $\ell_2$ diameter of $\mathcal{X}$,*

$$\langle F(\boldsymbol{x}^*), \boldsymbol{x} - \boldsymbol{x}^* \rangle \geq -2D\sqrt{L\epsilon} \quad \forall \boldsymbol{x} \in \mathcal{X}. \tag{15}$$

*Proof.* Let $\boldsymbol{x} \in \mathcal{X}$ be an arbitrary target point. For any $\delta \in (0, 1]$, we define the interpolation point $\boldsymbol{x}' = \boldsymbol{x}^* + \delta(\boldsymbol{x} - \boldsymbol{x}^*) \in \mathcal{X}$. Using the fact that $\boldsymbol{x}^*$ is an $\epsilon$-MVI solution,

$$\delta \langle F(\boldsymbol{x}'), \boldsymbol{x} - \boldsymbol{x}^* \rangle \geq -\epsilon \implies \langle F(\boldsymbol{x}'), \boldsymbol{x} - \boldsymbol{x}^* \rangle \geq -\frac{\epsilon}{\delta}.$$

We now relate $F(\boldsymbol{x}')$ to $F(\boldsymbol{x}^*)$ using the Lipschitz continuity of $F$:

$$
\begin{aligned}
\langle F(\boldsymbol{x}^*), \boldsymbol{x} - \boldsymbol{x}^* \rangle &= \langle F(\boldsymbol{x}'), \boldsymbol{x} - \boldsymbol{x}^* \rangle + \langle F(\boldsymbol{x}^*) - F(\boldsymbol{x}'), \boldsymbol{x} - \boldsymbol{x}^* \rangle \\
&\geq -\frac{\epsilon}{\delta} - \|F(\boldsymbol{x}^*) - F(\boldsymbol{x}')\|_2 \|\boldsymbol{x} - \boldsymbol{x}^*\|_2 \\
&\geq -\frac{\epsilon}{\delta} - L\|\boldsymbol{x}^* - \boldsymbol{x}'\|_2 \|\boldsymbol{x} - \boldsymbol{x}^*\|_2 \\
&= -\frac{\epsilon}{\delta} - L\delta\|\boldsymbol{x} - \boldsymbol{x}^*\|_2^2 \\
&\geq -\frac{\epsilon}{\delta} - L\delta D^2.
\end{aligned}
$$

The claim follows by picking $\delta$ optimally. $\qquad\square$

We now extend this approach under *hypomonotonicity* (Iusem et al., 2003; Alber et al., 2005; Alomar & Chavdarova, 2024). In particular, a mapping $F$ satisfies $\sigma$-hypomonotonicity for $\sigma > 0$ if

$$\langle F(\boldsymbol{x}) - F(\boldsymbol{x}'), \boldsymbol{x} - \boldsymbol{x}' \rangle \geq -\sigma\|\boldsymbol{x} - \boldsymbol{x}'\|^2 \tag{16}$$

for all $\boldsymbol{x}, \boldsymbol{x}' \in \mathcal{X}$. Starting again from (13), we have that for any $\boldsymbol{x}' \in \mathcal{X}$,

$$
\begin{aligned}
\epsilon &\geq \mathbb{E}_{\boldsymbol{x} \sim \mu}\left[\langle F(\boldsymbol{x}'), \boldsymbol{x} - \boldsymbol{x}' \rangle - \sigma\|\boldsymbol{x} - \boldsymbol{x}'\|^2\right] \\
&= \langle F(\boldsymbol{x}'), \overline{\boldsymbol{x}} - \boldsymbol{x}' \rangle - \sigma\mathbb{E}_{\boldsymbol{x} \sim \mu}[\|\boldsymbol{x} - \boldsymbol{x}'\|^2].
\end{aligned}
$$

As a result,

$$\langle F(\boldsymbol{x}'), \boldsymbol{x}' - \overline{\boldsymbol{x}} \rangle \geq -\epsilon - \sigma D^2 \quad \forall \boldsymbol{x}' \in \mathcal{X}.$$

This means that $\overline{\boldsymbol{x}}$ is an $(\epsilon + \sigma D^2)$-approximate MVI solution. Combining with Lemma C.4, we have shown the following.

**Proposition C.5.** *Let $F : \mathcal{X} \to \mathbb{R}^d$ be a $\rho$-hypomonotone L-Lipschitz continuous operator. There is a $\mathsf{poly}(d, \log(1/\epsilon))$-time algorithm for computing an $(\epsilon + \sigma D^2)$-approximate MVI solution, which is in turn a $2D\sqrt{L(\epsilon + \sigma D^2)}$-approximate VI solution.*

We now show how to translate this result for finding fixed points of a slightly expansive mapping $T: \|T(\boldsymbol{x}) - T(\boldsymbol{x}')\|_2 \leq (1+\sigma)\|\boldsymbol{x} - \boldsymbol{x}'\|_2$. As in (11), if $F(\boldsymbol{x}) = \boldsymbol{x} - T(\boldsymbol{x})$, we have

$$\langle F(\boldsymbol{x}) - F(\boldsymbol{x}'), \boldsymbol{x} - \boldsymbol{x}' \rangle = \frac{1}{2} \left( \|\boldsymbol{x} - \boldsymbol{x}'\|_2^2 - \|T(\boldsymbol{x}) - T(\boldsymbol{x}')\|_2^2 + \|\boldsymbol{x} - T(\boldsymbol{x}) - \boldsymbol{x}' + T(\boldsymbol{x}')\|_2^2 \right)$$

$$\geq - \left( \sigma + \frac{\sigma^2}{2} \right) \|\boldsymbol{x} - \boldsymbol{x}'\|_2^2$$

for any $\boldsymbol{x}, \boldsymbol{x}' \in \mathcal{X}$. In other words, $F$ is $(\sigma + \frac{\sigma^2}{2})$-hypomonotone. Furthermore, if $\boldsymbol{x}^*$ is an $\epsilon'$-VI solution for $F$, it follows that $\|T(\boldsymbol{x}^*) - \boldsymbol{x}^*\|_2 \leq \sqrt{\epsilon'}$. We arrive at the following conclusion.

**Proposition C.6.** *Let $T : \mathcal{X} \to \mathcal{X}$ be a such that $\|T(\boldsymbol{x}) - T(\boldsymbol{x}')\|_2 \leq (1+\sigma)\|\boldsymbol{x} - \boldsymbol{x}'\|_2$. There is a $\mathsf{poly}(d, \log(1/\epsilon))$-time algorithm for computing an $\epsilon'$-fixed point of $T$, where*

$$\epsilon' = \sqrt{2D\sqrt{(2+\sigma)\left( \epsilon + \left( \sigma + \frac{\sigma^2}{2} \right) D^2 \right)}}.$$

*In particular, if $\sigma \leq \epsilon$, $\epsilon' = \Theta_\epsilon(\epsilon^{1/4})$.*

Compared to the recent result of Diakonikolas (2025), the complexity above grows logarithmically in $1/\epsilon$, at the cost of being applicable to a narrower regime of $\rho$. Furthermore, as we highlighted in Section 3, Proposition C.5 yields a polynomial-time algorithm for computing $\epsilon$-performatively stable points in the following regime.

**Theorem 3.5.** *If $\rho \leq 1 + \epsilon$ (per Assumption 2.3), there is a $\mathsf{poly}(d, \log(1/\epsilon))$-time algorithm for computing an $O_\epsilon(\epsilon^{1/4})$-performatively stable point.*

Whether the tradeoff between approximation and expansiveness can be improved to match the result of Diakonikolas (2025) is an interesting question. As becomes evident from Propositions C.5 and C.6, the $\epsilon^{1/4}$ factor is an artifact of how approximation is measured. In terms of the VI problem corresponding to $F(\boldsymbol{x}) = \boldsymbol{x} - T(\boldsymbol{x})$, our approach yields an $O_\epsilon(\epsilon)$ approximation for a Minty VI solution and an $O_\epsilon(\sqrt{\epsilon})$ approximation for a (Stampacchia) VI solution.

# D. PPAD-hardness for general convex sets

In this section, we generalize the result of Theorem 3.4 from the domain $[0,1]^d$ to general convex sets. Through out the section, we let $n$ denote the bit-length of the input to the Turing machine. We start this section by defining the 2D-SPERNER problem. Consider the triangle $\triangle A_1 A_2 A_3$ on a 2D plane where $A_0 = (0,0)$, $A_1 = (2^n, 0)$, and $A_2 = (0, 2^n)$. We define the triangulation to be

$$T_n = \{ \boldsymbol{p} = (p_1, p_2) \in \mathbb{Z}^2 \mid p_1 \geq 0, p_2 \geq 0, p_1 + p_2 \leq 2^n \}.$$

For any 3-coloring function $g : T_n \to \{1, 2, 3\}$, it is said to be admissible if the following conditions are met:

- $g(A_i) = i$, for all $i \in \{1, 2, 3\}$;

- For every $\boldsymbol{p}$ on the segment of $A_i A_j$, $g(\boldsymbol{p}) \neq 6 - i - j$.

**Definition D.1** (2D-SPERNER ; Papadimitriou, 1990)**.** Given a polynomial-time Turing machine $F$ that produces a admissible 3-coloring $g$ on $T_n$ where $g(\boldsymbol{p}) = F(\boldsymbol{p}) \in \{1, 2, 3\}$ for every $\boldsymbol{p} \in T_n$, the output of 2D-SPERNER is a trichromatic triangle of coloring $g$.

The PPAD-membership of 2D-SPERNER was established by Papadimitriou (1990), Chen & Deng (2009) showed that 2D-SPERNER is PPAD-complete.

**Theorem D.2** (Chen & Deng, 2009)**.** 2D-SPERNER *is* PPAD-*complete.*

We note that even though the PPAD-hardness result for 2D-SPERNER is established on a right triangle, one can generalize this hardness result to arbitrary triangles.

**Lemma D.3.** *For any triangle $\triangle A_1 A_2 A_3$ where $A_1 = (0,0), A_2 = (a_1, a_2) = \boldsymbol{a}, A_3 = (b_1, b_2) = \boldsymbol{b}$, define the triangulation to be*

$$\mathcal{T}_n = \left\{ \boldsymbol{p} = \frac{q}{2^n}\boldsymbol{a} + \frac{r}{2^n}\boldsymbol{b} \mid (q,r) \in \mathbb{Z}^2, q \geq 0, r \geq 0, q + r \leq 2^n \right\}.$$

*Given a polynomial-time Turing machine $F'$ that produces an admissible 3-coloring $g'$ for all points $\boldsymbol{p} \in \mathcal{T}_n$, it is* PPAD-*complete to output a trichromatic triangle of coloring $g'$.*

*Proof.* First, observe that given $\boldsymbol{p}, \boldsymbol{a}$ and $\boldsymbol{b}$, we can compute coefficients $q$ and $r$ in polynomial time through standard basis decomposition. The PPAD-membership follows from Sperner's lemma. To prove the hardness, given a 2D-SPERNER instance $(F, 0^n)$, we construct the coloring $g'$ of triangle $\triangle A_1 A_2 A_3$ such that for any point $\boldsymbol{p} = (\frac{q}{2^n}\boldsymbol{a}, \frac{r}{2^n}\boldsymbol{b}) \in \mathcal{T}_n$

$$g'(\boldsymbol{p}) = F((q,r)).$$

Since $F$ produces an admissible 3-coloring, it holds that

- $g'(A_i) = i$, for all $i \in \{1, 2, 3\}$;

- For every $\boldsymbol{p} = (\frac{q}{2^n}\boldsymbol{a}, \frac{r}{2^n}\boldsymbol{b})$ on the segment of $A_i A_j, g'(\boldsymbol{p}) = g((q,r)) \neq 6 - i - j$.

Thus we show that $g'$ is an admissible 3-coloring over the triangulation $\mathcal{T}_n$. Furthermore, from any trichromatic triangle of coloring $g'$ over $\mathcal{T}_n$, we can recover a trichromatic triangle of coloring $g$ in $T_n$ in polynomial time. From Theorem D.2, we conclude the problem is PPAD-complete. $\square$

We now introduce the problem of $\epsilon$-THICKBROUWER (Deligkas et al., 2020), which is a extension of 2D-SPERNER on an arbitrary triangle $\triangle A_1 A_2 A_3$ such that the coloring $g(\cdot)$ satisfies the following boundary conditions:

For a given $\epsilon$ and any $\boldsymbol{p} = \frac{q}{2^n}\boldsymbol{a} + \frac{r}{2^n}\boldsymbol{b} \in \mathcal{T}_n$, it holds that

$$g(\boldsymbol{p}) = \begin{cases} 1 & \text{for all } q \leq 2^n \epsilon, \text{ and for all } 2^n \epsilon < r < (1 - \epsilon)2^n - q; \\ 2 & \text{for all } r \leq 2^n \epsilon, \text{ and for all } q < (1 - \epsilon)2^n - r; \\ 3 & \text{for all } q \text{ and } r \text{ such that } (1 - \epsilon)2^n \leq q + r \leq 2^n; \\ \text{any color in } \{1, 2, 3\} & \text{otherwise.} \end{cases} \tag{17}$$

Given a 2D-SPERNER instance, one can reduce it to $\epsilon$-THICKBROUWER in polynomial time by increasing the number of points in the triangle and embedding the original instance in the center of the new construction. A detailed proof can be found in the paper of Deligkas et al. (2020).

To map the coloring defined on the grid $\mathcal{T}_n$ to the triangle $\triangle A_1 A_2 A_3$, we adopt the bit-extraction technique, which is commonly used in PPAD-reductions. Specifically, consider a triangle $\triangle A_1 A_2 A_3$ with vertices $A_1 = \boldsymbol{0}$, $A_2 = \boldsymbol{a}$, and $A_3 = \boldsymbol{b}$. For any point $\boldsymbol{p}$ inside the triangle $\triangle A_1 A_2 A_3$, we can compute coefficients $q$ and $r$ through standard basis decomposition such that

$$\boldsymbol{p} = \frac{q}{2^n}\boldsymbol{a} + \frac{r}{2^n}\boldsymbol{b}.$$

---

**Algorithm 1** ExtractBit $(x, b)$

---

$\quad b \leftarrow 0.5$
$\quad b \leftarrow x -^b b$
$\quad b \leftarrow b *^b L$

---

We then apply the bit-extraction scheme Algorithm 1 of Deligkas et al. (2020) to recover the first $n$ bits of $\frac{q}{2^n}$ and $\frac{r}{2^n}$. Operators $+^b, -^b, *^b$ denote bounded operations that ensure the outcomes remain in $[0, 1]$, which can be efficiently implemented through a algorithmic circuit with standard $\min$ and $\max$ operations. Notice that when $x \leq 0.5$, we have $b = 0$, and when $x \geq 0.5 + \frac{1}{L}$, we have $b = 1$. For $0.5 < x < 0.5 + \frac{1}{L}$, the value of $b$ may lie anywhere strictly between 0 and 1 due to the continuity of the output of the algorithmic circuit. We refer to the first two cases as well-positioned and the last

case as poorly-positioned. To account for the effect of poorly-positioned points, for any $x \in [0,1]^2$, we sample $k$ points $x_1, x_2, \ldots, x_k$ where

$$x_i = x + (i-1)\left[\frac{1}{(k+1)2^{n+1}}, \frac{1}{(k+1)2^{n+1}}\right].$$

The following lemma holds for the sample points $x_1, \ldots, x_k$.

**Lemma D.4** (Deligkas et al., 2020). *Setting $L = (k+2)2^{n+1}$, then among points $x_1, \ldots, x_k$, at most two points will be poorly-positioned.*

We proceed to restate the main results of this section.

**Theorem 3.12.** *Given a convex and compact domain $\mathcal{X} \subset \mathbb{R}^d$ that is well bounded, an $L$-Lipschitz function $F : \mathcal{X} \to \mathbb{R}^d$, and $\epsilon = O(2^{-n})$, it is PPAD-hard to find a point $x^* \in \mathcal{X}$ such that*

$$\langle x - x^*, F(x^*) \rangle \leq \epsilon \quad \forall x \in \mathcal{X}. \tag{4}$$

*This holds even when $d = 2$ and $L = O(1)$.*

*Proof.* Let $\mathcal{X} \subset \mathbb{R}^2$ be a two-dimensional well-bounded domain. From Definition 3.11, there exist a 2D ball $\mathcal{B}_{R_1}$ inside $\mathcal{X}$. Consider an arbitrary triangle equilateral triangle $\triangle A_1 A_2 A_3$ that lies on the boundary of $\mathcal{B}_{R_1}$. Without loss of generality, we assume $R_1 = 1$ and further assume the position of $A_1$ is at $(0,0)$, $A_2 = (\sqrt{3}, 0) = a$, and $A_3 = (\frac{\sqrt{3}}{2}, \frac{3}{2}) = b$. Let the discretized grid over $\triangle A_1 A_2 A_3$ be as defined in Lemma D.3. We set $\epsilon = \frac{1}{8}$ and assign colors to the grid points according to (17) such that the coloring $g(\cdot)$ for points on the grid $\mathcal{T}_n$ is admissible for the $\epsilon$-THICKBROUWER problem. For point $x \in \mathcal{X}$ outside $\triangle A_1 A_2 A_3$, the coloring $g(x)$ is defined as

$$g(x) = \begin{cases} 1 \text{ if } \min\{\text{dist}(x, A_1 A_2), \text{dist}(x, A_1 A_3), \text{dist}(x, A_2 A_3)\} = \text{dist}(x, A_1 A_3); \\ 2 \text{ if } \min\{\text{dist}(x, A_1 A_2), \text{dist}(x, A_1 A_3), \text{dist}(x, A_2 A_3)\} = \text{dist}(x, A_1 A_2); \\ 3 \text{ if } \min\{\text{dist}(x, A_1 A_2), \text{dist}(x, A_1 A_3), \text{dist}(x, A_2 A_3)\} = \text{dist}(x, A_2 A_3); \\ \text{In terms of ties:} \\ 1 \text{ if } \min\{\text{dist}(x, A_1 A_2), \text{dist}(x, A_1 A_3), \text{dist}(x, A_2 A_3)\} = \text{dist}(x, A_1 A_2) = \text{dist}(x, A_1 A_3); \\ 2 \text{ if } \min\{\text{dist}(x, A_1 A_2), \text{dist}(x, A_1 A_3), \text{dist}(x, A_2 A_3)\} = \text{dist}(x, A_1 A_2) = \text{dist}(x, A_2 A_3); \\ 3 \text{ if } \min\{\text{dist}(x, A_1 A_2), \text{dist}(x, A_1 A_3), \text{dist}(x, A_2 A_3)\} = \text{dist}(x, A_1 A_3) = \text{dist}(x, A_2 A_3), \end{cases} \tag{18}$$

where $\text{dist}(x, A_i A_j)$ denotes the distance from point $x$ to line $A_i A_j$. Notice that by construction, there is no trichromatic triangle outside $\triangle A_1 A_2 A_3$.

We move on to map the coloring of $x$ to the operator value $F(x)$. Recall that segment $A_1 A_2 = a$ and segment $A_1 A_3 = b$, we define $a_\perp = (0,1)$ to be the unit vector orthogonal to $a$ and pointing inside the triangle. Similarly, we define $b_\perp = (\frac{\sqrt{3}}{2}, -\frac{1}{2})$ and $c_\perp = (-\frac{\sqrt{3}}{2}, -\frac{1}{2})$ as the unit vectors orthogonal to segment $A_1 A_3$ and $A_2 A_3$ respectively, each pointing inward, as shown in Figure 2. Since $\triangle A_1 A_2 A_3$ is a equilateral triangle, it follows that $a_\perp + b_\perp + c_\perp = 0$. We then map each color to a different vector such that color 1 is mapped to $b_\perp$, color 2 is mapped to $a_\perp$, and color 3 is mapped to $c_\perp$. For any point $x \in \mathcal{X}$, we first sample $k$ points[2] such that

$$x_i = x + (i-1)\left[\frac{1}{(k+1)2^{n+1}}a, \frac{1}{(k+1)2^{n+1}}b\right].$$

We then extract the first $n$ bits of $x_i$ as $\overline{x}_i$ and pass to the boolean circuit to get the corresponding color. The operator $F(x)$ is then computed as the average of the vectors corresponding to the colors of sampled points,

$$F(x) = \frac{1}{k}\left(\sum_{i=1}^{k} \mathbb{1}(g(\overline{x}_i) = 2)a_\perp + \mathbb{1}(g(\overline{x}_i) = 1)b_\perp + \mathbb{1}(g(\overline{x}_i) = 3)c_\perp\right),$$

where $\mathbb{1}(g(\overline{x}_i) = j)$ denotes the indicator that the coloring of $\overline{x}_i$ is $j$.

---

[2]The coloring for any sample point $x_i$ outside $\mathcal{X}$ is also determined by (18).

What remains now is to show that from a solution $x^*$ of (4), one can recover a trichromatic triangle. We begin by showing that any point within $\epsilon$ distance from the boundary of $\mathcal{X}$ cannot be a solution of (4).

If $x$ lies within $\frac{\epsilon}{2}$ distance of the boundary of $\mathcal{X}$, it either lies outside of $\triangle A_1 A_2 A_3$ or is within $\frac{\epsilon}{2}$ distance from one of the segment $A_i A_j$. By the coloring rule in (17) and (18), among the sampled points $x_1, \ldots x_k$, any well-positioned point can only take two of the three colors. Without loss of generality, we assume that color 2 is missing from all well-positioned sample points of $x$, the cases where color 1 or color 3 is missing follow similarly.

From Lemma D.4, at least $k-2$ points out of the $k$ sample points are well-conditioned and are assigned either color 1 or 3. We consider two cases,

- **Color** 1 **is also missing among the well-conditioned sample points**. By the coloring rule in (17) and the the choice of $\epsilon$, it follows that $x$ must lie within distance $\frac{\epsilon}{2}$ of segment $A_2 A_3$. Recall that $c_\perp = (-\frac{\sqrt{3}}{2}, -\frac{1}{2})$, we consider the $x$-coordinate of $F(x)$,

$$F(x)_x \leq -\left(\frac{\sqrt{3}(k-2)}{2k} - \frac{2}{k}\right),$$

where the first term comes from the contribution of the $k-2$ well-conditioned sample points, while the second term accounts for the error introduced by the remaining two points. Let $x' = A_1 = (0,0)$, we have

$$\langle x' - x, F(x) \rangle \geq (x'_x - x_x) F(x)_x$$
$$\geq \left(\frac{\sqrt{3}}{2} - \frac{\epsilon}{2}\right) \cdot \left(\frac{\sqrt{3}(k-2)}{2k} - \frac{2}{k}\right)$$
$$\geq \epsilon',$$

where the second inequality holds because $x$ is within $\epsilon$ distance from $A_2 A_3$ segment, so its $x$-coordinate, $x_x \geq \frac{\sqrt{3}}{2} - \epsilon$. The third inequality holds by setting $k \geq 16$ and $\epsilon' \leq \frac{\epsilon}{8} = \frac{1}{32}$.

- **The well-conditioned sample points contain both color** 1 **and color** 3. From the coloring rule in (17), $x$ cannot lie within $\frac{\epsilon}{2}$ distance with segment $A_1 A_2$. Recall that $b_\perp = (\frac{\sqrt{3}}{2}, -\frac{1}{2})$ and $c_\perp = (-\frac{\sqrt{3}}{2}, -\frac{1}{2})$. Consequently, the averaged direction $F(x)$ also has negative $y$-component. Specifically, let $F(x)_y$ denote the $y$-coordinate of $F(x)$, it holds that

$$F(x)_y \leq -\left(\frac{k-2}{2k} - \frac{2}{k}\right).$$

Let $x'$ be a point on the segment $A_1 A_2$. Note that since color 1 is missing, by the construction of the $\epsilon$-THICKBROUWER problem, we also have $x_y - x'_y \geq \frac{\epsilon}{2}$. Therefore

$$\langle x' - x, F(x) \rangle \geq (x'_y - x_y) F(x)_y$$
$$\geq \frac{\epsilon}{2} \cdot \left(\frac{k-2}{2k} - \frac{2}{k}\right)$$
$$> \epsilon',$$

where the last step follows by setting $k \geq 16$ and $\epsilon' \leq \frac{\epsilon}{8} = \frac{1}{32}$.

For any point $x$ that lies more than $\frac{\epsilon}{2}$ distance away from the boundary, we argue that if $x$ is a solution for (4), then one can recover a trichromatic triangle for the $\epsilon$-THICKBROUWER problem. We show this by contradiction, first assume that if color 1 is missing from the well-positioned sampled points among $x_1 \cdots x_k$, like the previous case, we have

$$F(x)_y \leq -\left(\frac{k-2}{2k} - \frac{2}{k}\right).$$

Note that since $x$ is not within $\frac{\epsilon}{2}$ distance from the boundary, along the negative $y$ direction, we can find a point $x' \in \mathcal{X}$ that is at least $\epsilon$ away from $x$ (i.e., $x' = x - \epsilon \cdot (0, 1)$). It then holds that

$$\langle x' - x, F(x) \rangle > \frac{\epsilon}{2} \cdot \left(\frac{k-2}{2k} - \frac{2}{k}\right) > \epsilon'.$$

The cases where color 2 or color 3 is missing follow similarly. Therefore we conclude that if $x$ is a solution of (4), the well-positioned sample points among $x_1, \ldots, x_n$ must have all three colors. Observe that $\|x_k - x\|_\infty < \frac{1}{2^n} \min(\|a\|_2, \|b\|_2)$, which implies that if the well-positioned points among $x_1 \ldots x_k$ contain all three colors, then $x$ must reside within a trichromatic square with sides oriented along directions $a$ and $b$. Such a square can only occur within $\triangle A_1 A_2 A_3$. Finally, the reduction from trichromatic triangles to trichromatic squares of 2D-SPERNER is established in Chen & Deng (2009).

Note that since the grid $\mathcal{T}_n$ has side length $O(\frac{1}{2^n})$, the Lipschitz constant of the operator $F(\cdot)$ is $O(2^n)$. Define the rescaled operator $F'(x) = \frac{F(x)}{2^n}$, and let $\epsilon'' = \frac{\epsilon'}{2^n} = O(\frac{1}{2^n})$. Then computing a point $x^* \in \mathcal{X}$ such that for any $x \in \mathcal{X}$,

$$\langle x - x^*, F'(x^*) \rangle \leq \epsilon''$$

is PPAD-hard. Moreover, the Lipschitz constant for $F'(\cdot)$ is $O(1)$ and $\epsilon'' = O(\frac{1}{2^n})$. This completes the proof. $\qquad\square$

*Remark* D.5. In our proof the operator $F$ is given by a (well-behaved) arithmetic circuit with $n$ rational inputs and size that depends polynomially on the description of the 2D-SPERNER problem, which can effectively approximate any Lipschitz continuous function. We refer the reader to Fearnley et al. (2023) for further background on complexity theory.

*Remark* D.6. We remark that our construction uses irrational coordinates for the positions of $A_2 \, A_3$ and for the directional vectors $a_\perp, b_\perp, \, c_\perp$, which cannot be represented exactly by a Turing machine. Nevertheless, our reduction continues to work given a suitably good approximation of these quantities. A similar technical issue is discussed in Deligkas et al. (2020).

## E. PLS-hardness of finding local optima in strategic classification

In this section, we establish that finding local optima in strategic classification per Definition 4.1 is PLS-hard. We first restate the main result we want to prove.

**Theorem 4.4.** *Given a finite population $X$, a distribution $\mathcal{D}$ over $X$, a cost function $c$, and a target classifier $h$, it is* PLS-*hard to find a strategic local optimum as in Definition 4.3. This result holds even when $c$ is a metric and the target classifier $h$ is provided explicitly to the algorithm.*

*Proof.* The proof proceeds via a polynomial-time reduction from the LOCALMAXCUT problem. Let $G = (V, E, w)$ be a weighted undirected graph with edge weights $w_{(u,v)} \geq 0$ for any edge $(u, v) \in E$. We construct a strategic classification instance with a finite population $X$ and a non-uniform distribution $\mathcal{D}$ over $X$. For convenience, we define the *weight* of a point $w_{\mathcal{D}}(x)$ so that the probability of sampling $x \in X$ from $\mathcal{D}$ is proportional to $w_{\mathcal{D}}(x)$. The population is defined as follows.

- For each vertex $v \in V$, we introduce a point $x_{v^-}$ with label $h(x_{v^-}) = 0$. The weight of $x_{v^-}$ under distribution $\mathcal{D}$ is given by the total weight of edges incident to vertex $v$, i.e., $w_{\mathcal{D}}(x_{v^-}) = \sum_{u \in \mathcal{N}(v)} w_{(u,v)}$;

- For every edge $(u, v) \in E$, we introduce a point $x_{(u,v)^+}$ with label $h(x_{(u,v)^+}) = 1$, and weight $w_{\mathcal{D}}(x_{(u,v)^+}) = 2w_{(u,v)}$;

- For every edge $(u, v) \in E$, we introduce a point $x_{(u,v)^-}$ with label $h(x_{(u,v)^-}) = 0$, and weight $w_{\mathcal{D}}(x_{(u,v)^-}) = 2w_{(u,v)} + 1$.

We now define a metric $c : X \times X \to \mathbb{R}_{\geq 0}$. We choose the value such that $c(x, x) = 0$ and $c(x, y) = c(y, x)$. Moreover, the metric $c$ takes only two nonzero values, $0.8$ and $1.2$. These values are chosen to ensure that the triangle inequality holds, other than that, we can set them to arbitrary values in the range from $(0, 1)$ and $(1, \infty)$ respectively. The metric is defined as follows.

- For each vertex point $x_{v^-}$, and for every positive edge point $x_{(u,v)^+}$ such that edge $(u, v)$ is incident to vertex $v$, we set $c(x_{v^-}, x_{(u,v)^+}) = 0.8$;

- For each edge positive point $x_{(u,v)^+}$ and corresponding edge negative point $x_{(u,v)^-}$ we set $c(x_{(u,v)^+}, x_{(u,v)^-}) = 0.8$;

- For all other pairs $(x, y)$, we define $c(x, y) = 1.2$.

If $c(x, y) = 0.8$, we call them close to each other. Notice that under this metric $c$, the Contestant will only deviate a point $x$ to point $y$ if $f(x) = 0$, $f(y) = 1$, and $x$ is close to $y$.

The first claim is that if $f^*$ is at a strategic local optimum, then $f^*(x_{(u,v)-}) = 0$ and $f^*(x_{(u,v)+}) = 0$ for all edge $(u, v) \in E$. To see this, assume that $f^*(x_{(u,v)-}) = 1$ for some edge $(u, v) \in E$, then the Jury can simply deviate to another classifier $f'$ that differs with $f^*$ with only the prediction of $x_{(u,v)-}$. If $M$ is the sum of all the weights of points in $X$, it holds that

$$\Pr_{\boldsymbol{x} \sim \mathcal{D}}[h(\boldsymbol{x}) = f'(\Delta(\boldsymbol{x}))] - \Pr_{\boldsymbol{x} \sim \mathcal{D}}[h(\boldsymbol{x}) = f^*(\Delta(\boldsymbol{x}))] \geq \frac{1}{M}\left(2w_{(u,v)} + 1 - 2w_{(u,v)}\right) \tag{19}$$

$$= \frac{1}{M}, \tag{20}$$

where (19) holds because changing the prediction of $x_{(u,v)-}$ from 1 to 0 may cause the misclassification of the positive edge point $x_{(u,v)+}$, but it ensures that the negative edge point $x_{(u,v)-}$ is classified correctly. As a result, (20) implies that the Jury can strictly improve their utility by deviating to the classifier $f'$, which contradicts the assumption that $f^*$ is a strategic local optimum.

Now suppose $f^*(x_{(u,v)+}) = 1$ for some $(u, v) \in E$. Since $f^*(x_{(u,v)-}) = 0$ and $x_{(u,v)+}$ and $x_{(u,v)-}$ are close to each other, the Contestant will deviate $x_{(u,v)-}$ to $x_{(u,v)+}$. Consider an alternative classifier $f'$ that differs from $f^*$ only in the prediction of $x_{(u,v)+}$, we have

$$\Pr_{\boldsymbol{x} \sim \mathcal{D}}[h(\boldsymbol{x}) = f'(\Delta(\boldsymbol{x}))] - \Pr_{\boldsymbol{x} \sim \mathcal{D}}[h(\boldsymbol{x}) = f^*(\Delta(\boldsymbol{x}))] \geq \frac{1}{M}\left(-2w_{(u,v)} - (-2w_{(u,v)} - 1)\right)$$

$$= \frac{1}{M},$$

where the first inequality holds because changing the prediction of $x_{(u,v)+}$ from 1 to 0 may cause the positive edge point $x_{(u,v)+}$ to be misclassified, but it ensures that the negative edge point $x_{(u,v)-}$ is classified correctly. Moreover, any deviations of vertex points $x_{v-}$ can only increase this gap. Thus, $f^*$ cannot be a strategic local optimum, yielding a contradiction.

We conclude that, in order to reach a strategic local optimum, the only points that can be labeled positively are the vertex points $x_{v-}$. In this case, the Jury can improve their utility through the deviation of positive edge points $x_{(u,v)+}$ to the corresponding vertex points $x_{v-}$.

We now proceed to analyze the utility of the Jury when the label of a single vertex point $x_{v-}$ is changed from 0 to 1. Let $\mathcal{N}(v)^+$ denote the set of neighbors of vertex $v$ in the original graph whose corresponding vertex points are labeled 1 by the classifier, and let $\mathcal{N}(v)^-$ denote the set of neighbors that are labeled 0. Observe that before changing the label of $x_{v-}$, for all vertices $u \in \mathcal{N}(v)^+$, the Contestant already deviates the corresponding positive edge points $x_{(u,v)+}$ to $x_{u-}$. Hence, those points will be correctly labeled regardless of the change. In contrast, for every $u \in \mathcal{N}(v)^-$, the corresponding positive edge points $x_{(u,v)+}$ do not deviate before the change, but will deviate to $x_{v-}$ after the change.

Let $f$ denote the Jury's classifier before the change and $f'$ the classifier after the change. The resulting difference in the Jury's utility is

$$\Pr_{\boldsymbol{x} \sim \mathcal{D}}[h(\boldsymbol{x}) = f'(\Delta(\boldsymbol{x}))] - \Pr_{\boldsymbol{x} \sim \mathcal{D}}[h(\boldsymbol{x}) = f(\Delta(\boldsymbol{x}))] = \frac{1}{M}\left(\sum_{u \in \mathcal{N}(v)^-} 2w_{(u,v)} - \sum_{u' \in \mathcal{N}(v)} w_{(u',v)}\right) \tag{21}$$

$$= \frac{1}{M}\left(\sum_{u \in \mathcal{N}(v)^-} w_{(u,v)} - \sum_{u' \in \mathcal{N}(v)^+} w_{(u',v)}\right). \tag{22}$$

The first term in (21) corresponds to the gain from the correctly labeling positive edge points after the change, while the second term accounts for the loss introduced by misclassifying the vertex point $x_{v-}$.

Similarly, consider the case where the Jury change the label of a vertex point $x_{v-}$ from 1 to 0. For each vertex $u \in \mathcal{N}(v)^+$, the corresponding edge points $x_{(u,v)+}$ will still be classified positive since the Contestant will deviate to $x_{u-}$. However, for every vertex $u \in \mathcal{N}(v)^-$, edge points $x_{(u,v)+}$ will be misclassified, since after the change they no longer have any positively labeled neighbors. Thus, the resulting change in the Jury's utility is

$$\Pr_{\boldsymbol{x} \sim \mathcal{D}}[h(\boldsymbol{x}) = f'(\Delta(\boldsymbol{x}))] - \Pr_{\boldsymbol{x} \sim \mathcal{D}}[h(\boldsymbol{x}) = f(\Delta(\boldsymbol{x}))] = \frac{1}{M}\left(-\sum_{u \in \mathcal{N}(v)^-} 2w_{(u,v)} + \sum_{u' \in \mathcal{N}(v)} w_{(u',v)}\right) \quad (23)$$

$$= \frac{1}{M}\left(\sum_{u \in \mathcal{N}(v)^+} w_{(u,v)} - \sum_{u' \in \mathcal{N}(v)^-} w_{(u',v)}\right). \quad (24)$$

Here, the first term in (23) captures the loss from misclassifying edge points $x_{(u,v)^+}$, while the second term is due to correctly labeling the vertex point $x_{v^-}$.

Suppose we have a classifier $f^*$ which is at a strategic local optimum. By Definition 4.3, (22) is nonpositive for every vertex $u$ such that $x_{u^-}$ is labeled 0, and (24) is nonpositive for every vertex $v$ such that $x_{v^-}$ is labeled 1. Now consider a cut of the original graph defined as follows: each vertex $v \in V$ with $f^*(x_{v^-}) = 0$ is on one side of the cut (negative side), and each vertex with $f^*(x_{v^-}) = 1$ is placed on the other side (positive side). Since (22) is nonpositive, moving any vertex from the negative side of the cut to the positive side cannot increase the total weight of the cut. Similarly, since (24) is nonpositive, moving any vertex from the positive side to the negative side also cannot improve the cut weight. Thus, we conclude that any locally strategic optimal classifier $f^*$ induces a local max cut on the original graph $G$. This completes the proof. $\qquad \square$

## F. Further omitted proofs

This section contains additional omitted proofs. We begin with Theorem 3.4.

**Theorem 3.4.** *Finding an $\epsilon$-performatively stable point per Definition 2.4 is PPAD-hard even when $L\beta/\alpha \leq 1 + \frac{\epsilon}{\epsilon'}$ for $\epsilon' = 0.088/6 \approx 0.0147$. This is so even when $\ell$ is a quadratic objective, $\ell(\boldsymbol{x}; \boldsymbol{z}) = \frac{1}{2}\|\boldsymbol{x} - \boldsymbol{z}\|_2^2$, and $\mathcal{D}(\boldsymbol{x})$ is given by an affine map.*

*Proof.* Let $\mathcal{X} = [0,1]^d$ and let $\boldsymbol{x}^* \in \mathcal{X}$ be an $\epsilon$-performatively stable point of (2)-(3). By Definition 2.4, we have that for all $\boldsymbol{x} \in \mathcal{X}$,

$$\langle \boldsymbol{x} - \boldsymbol{x}^*, \boldsymbol{x}^* - g(\boldsymbol{x}^*)\rangle \geq -\epsilon. \quad (25)$$

Now, let $g(\boldsymbol{x}) : (\mathbf{I} - \overline{\mathbf{A}})\boldsymbol{x} - \overline{\boldsymbol{b}}$, where $\overline{\mathbf{A}} = \frac{\epsilon}{\epsilon'}\mathbf{A}$ and $\overline{\boldsymbol{b}} = \frac{\epsilon}{\epsilon'}\boldsymbol{b}$ for $\mathbf{A}$ and $\boldsymbol{b}$ as in Lemma 3.3. Finding a solution $\boldsymbol{x}^*$ satisfying (25) would imply that for all $\boldsymbol{x} \in [0,1]^d$,

$$\langle \boldsymbol{x} - \boldsymbol{x}^*, \mathbf{A}\boldsymbol{x}^* + \boldsymbol{b}\rangle \geq -\epsilon'. \quad (26)$$

From Lemma 3.3, we conclude that it is PPAD-complete to find a point $\boldsymbol{x}^*$ satisfying (25). Furthermore,

$$\begin{aligned}
\|g(\boldsymbol{x}) - g(\boldsymbol{x}')\| &= \left\|(\mathbf{I} - \overline{\mathbf{A}})\boldsymbol{x} - (\mathbf{I} - \overline{\mathbf{A}})\boldsymbol{x}'\right\| \\
&\leq (\|\mathbf{I}\| + \|\overline{\mathbf{A}}\|_2)\|\boldsymbol{x} - \boldsymbol{x}'\| \\
&\leq (1 + \sqrt{\|\overline{\mathbf{A}}\|_1 \|\overline{\mathbf{A}}\|_\infty})\|\boldsymbol{x} - \boldsymbol{x}'\| \\
&\leq (1 + \frac{\epsilon}{\epsilon'})\|\boldsymbol{x} - \boldsymbol{x}'\|.
\end{aligned}$$

We conclude that even when $L\beta/\alpha \leq 1 + \frac{\epsilon}{\epsilon'}$, it is PPAD-complete to find an $\epsilon$-performatively stable point. $\qquad \square$

We next point out the polynomial equivalence between the two natural ways of measuring approximation for performatively stable points.

**Lemma F.1.** *If $\ell(\boldsymbol{x}; \boldsymbol{z})$ is $\alpha$-strongly convex in $\boldsymbol{x}$ (with respect to the $\|\cdot\|_2$ norm) for any $\boldsymbol{z} \in \mathcal{Z}$ and $\boldsymbol{x}^*$ is an $\epsilon$-performatively stable point (Definition 2.4), then*

$$\|\boldsymbol{x}^* - G(\boldsymbol{x}^*)\|_2 \leq \sqrt{\frac{\epsilon}{\alpha}},$$

*where $G$ is the RRM map (Definition 1.1).*

*Proof.* Since $\ell(\boldsymbol{x}; \boldsymbol{z})$ is $\alpha$-strongly convex in $\boldsymbol{x}$ for any $\boldsymbol{z}$, the expected loss $f(\boldsymbol{x}) = \mathbb{E}_{\boldsymbol{z} \sim \mathcal{D}(\boldsymbol{x}^*)}[\ell(\boldsymbol{x}; \boldsymbol{z})]$ is also $\alpha$-strongly convex. By strong convexity, we have for any $\boldsymbol{x}, \boldsymbol{x}' \in \mathcal{X}$,

$$\langle \nabla f(\boldsymbol{x}) - \nabla f(\boldsymbol{x}'), \boldsymbol{x} - \boldsymbol{x}' \rangle \geq \alpha \|\boldsymbol{x} - \boldsymbol{x}'\|_2^2. \tag{27}$$

We now write

$$\langle \nabla f(\boldsymbol{x}^*) - \nabla f(G(\boldsymbol{x}^*)), \boldsymbol{x}^* - G(\boldsymbol{x}^*) \rangle = \langle \nabla f(\boldsymbol{x}^*), \boldsymbol{x}^* - G(\boldsymbol{x}^*) \rangle - \langle \nabla f(G(\boldsymbol{x}^*)), \boldsymbol{x}^* - G(\boldsymbol{x}^*) \rangle.$$

We bound each term separately. First, since $\boldsymbol{x}^*$ is an $\epsilon$-performatively stable point, we have

$$\langle \boldsymbol{x} - \boldsymbol{x}^*, \nabla f(\boldsymbol{x}^*) \rangle \geq -\epsilon \quad \forall \boldsymbol{x} \in \mathcal{X}.$$

Taking $\boldsymbol{x} = G(\boldsymbol{x}^*)$, we get $\langle \nabla f(\boldsymbol{x}^*), \boldsymbol{x}^* - G(\boldsymbol{x}^*) \rangle \leq \epsilon$. Second, since $G(\boldsymbol{x}^*) = \operatorname{argmin}_{\boldsymbol{x} \in \mathcal{X}} f(\boldsymbol{x})$, the first-order optimality condition yields $\langle \boldsymbol{x} - G(\boldsymbol{x}^*), \nabla f(G(\boldsymbol{x}^*)) \rangle \geq 0$ for all $\boldsymbol{x} \in \mathcal{X}$, which in turn implies $-\langle \nabla f(G(\boldsymbol{x}^*)), \boldsymbol{x}^* - G(\boldsymbol{x}^*) \rangle \leq 0$. Combining these bounds with (27),

$$\alpha \|\boldsymbol{x}^* - G(\boldsymbol{x}^*)\|_2^2 \leq \langle \nabla f(\boldsymbol{x}^*) - \nabla f(G(\boldsymbol{x}^*)), \boldsymbol{x}^* - G(\boldsymbol{x}^*) \rangle \leq \epsilon,$$

and the proof follows. $\qquad \square$

**Lemma F.2.** *If $\ell(\boldsymbol{x}; \boldsymbol{z})$ satisfies $\|\nabla_{\boldsymbol{x}} \ell(\boldsymbol{x}; \boldsymbol{z}) - \nabla_{\boldsymbol{x}} \ell(\boldsymbol{x}'; \boldsymbol{z})\|_2 \leq \beta \|\boldsymbol{x} - \boldsymbol{x}'\|_2$, any point $\boldsymbol{x}^* \in \mathcal{X}$ such that $\|\boldsymbol{x}^* - G(\boldsymbol{x}^*)\|_2 \leq \epsilon$, where $G$ is the RRM map (Definition 1.1) is $\epsilon'$-performatively stable (Definition 2.4) with $\epsilon' = \epsilon \left( D\beta + \|\nabla f(G(\boldsymbol{x}^*))\|_2 \right)$, where $D$ is the $\ell_2$ diameter of $\mathcal{X}$.*

*Proof.* Let $f(\boldsymbol{x}) = \mathbb{E}_{\boldsymbol{z} \sim \mathcal{D}(\boldsymbol{x}^*)}[\ell(\boldsymbol{x}; \boldsymbol{z})]$. We have

$$\langle \boldsymbol{x} - \boldsymbol{x}^*, \nabla f(\boldsymbol{x}^*) \rangle = \langle \boldsymbol{x} - \boldsymbol{x}^*, \nabla f(G(\boldsymbol{x}^*)) \rangle + \langle \boldsymbol{x} - \boldsymbol{x}^*, \nabla f(\boldsymbol{x}^*) - \nabla f(G(\boldsymbol{x}^*)) \rangle. \tag{28}$$

For the first term, we write

$$\begin{aligned} \langle \boldsymbol{x} - \boldsymbol{x}^*, \nabla f(G(\boldsymbol{x}^*)) \rangle &= \langle \boldsymbol{x} - G(\boldsymbol{x}^*), \nabla f(G(\boldsymbol{x}^*)) \rangle + \langle G(\boldsymbol{x}^*) - \boldsymbol{x}^*, \nabla f(G(\boldsymbol{x}^*)) \rangle \\ &\geq \langle G(\boldsymbol{x}^*) - \boldsymbol{x}^*, \nabla f(G(\boldsymbol{x}^*)) \rangle, \end{aligned}$$

by the first-order optimality condition of $G(\boldsymbol{x}^*)$. Thus,

$$\langle \boldsymbol{x} - \boldsymbol{x}^*, \nabla f(G(\boldsymbol{x}^*)) \rangle \geq -\|\boldsymbol{x}^* - G(\boldsymbol{x}^*)\|_2 \|\nabla f(G(\boldsymbol{x}^*))\|_2 \geq -\epsilon \|\nabla f(G(\boldsymbol{x}^*))\|_2.$$

For the second term in the right-hand side of (28), we use $\beta$-Lipschitz continuity of $\nabla f$ to get

$$\langle \boldsymbol{x} - \boldsymbol{x}^*, \nabla f(\boldsymbol{x}^*) - \nabla f(G(\boldsymbol{x}^*)) \rangle \geq -\|\boldsymbol{x} - \boldsymbol{x}^*\|_2 \|\nabla f(\boldsymbol{x}^*) - \nabla f(G(\boldsymbol{x}^*))\|_2 \geq -\epsilon D\beta,$$

and the proof follows. $\qquad \square$

To conclude, we provide the proof of Proposition 4.5. We begin by stating a variation of Definition 4.1 that incorporates classifier-dependent costs. It also forces the Jury to select a classifier from a specified set, and similarly for the Contestant.

**Definition F.3** (Strategic classification with endogenous costs)**.** Strategic classification is a game played between the *Jury* and the *Contestant*. Let $\mathcal{D}$ be a distribution over a population $X$, $c : X \times X \to \mathbb{R}_{\geq 0}$ a cost function, and $h$ a target classifier.

1. The Jury first publishes a classifier $f_j : X \to \{0, 1\}$ selected from a set of classifiers $\{f_1, \ldots, f_m\}$.

2. The Contestant selects a deviation $\Delta_i : X \to X$ selected from a set of deviations $\{\Delta_1, \ldots, \Delta_n\}$.

The payoff to the Jury is $\Pr_{\boldsymbol{x} \sim \mathcal{D}}[h(\boldsymbol{x}) = f_j(\Delta_i(\boldsymbol{x}))]$ and the payoff to the Contestant is $\mathbb{E}_{\boldsymbol{x} \sim \mathcal{D}}[f_j(\Delta_i(\boldsymbol{x})) - c_j(\boldsymbol{x}, \Delta_i(\boldsymbol{x}))]$.

While the Jury has now to decide among a small set of possible classifiers (as opposed to $2^{|X|}$), we show that computing a performatively stable point is PPAD-hard.

**Proposition 4.5.** *Computing a performatively stable point in strategic classification with endogenous costs is* PPAD-*hard.*

Our reduction makes use of the hardness of Nash equilibria in two-player games.

**Definition F.4** (Nash equilibrium)**.** For a two-player game $(\mathbf{A}, \mathbf{B})$, with $\mathbf{A}, \mathbf{B} \in \mathbb{R}^{n \times m}$, an $\epsilon$-*Nash equilibrium* is a point $(\boldsymbol{x}, \boldsymbol{y}) \in \Delta^n \times \Delta^m$ such that

$$\langle \boldsymbol{x}, \mathbf{A}\boldsymbol{y} \rangle \geq \langle \hat{\boldsymbol{x}}, \mathbf{A}\boldsymbol{y} \rangle - \epsilon \text{ and } \langle \boldsymbol{x}, \mathbf{B}\boldsymbol{y} \rangle \geq \langle \boldsymbol{x}, \mathbf{B}\hat{\boldsymbol{y}} \rangle - \epsilon \quad \forall (\hat{\boldsymbol{x}}, \hat{\boldsymbol{y}}) \in \Delta^n \times \Delta^m.$$

*Proof of Proposition 4.5.* We reduce from the PPAD-hard problem of computing a Nash equilibrium of a win-loss game (Abbott et al., 2005). Let $(\mathbf{A}, \mathbf{B}) \in \{0, 1\}^{n \times m}$ be the payoff matrices for the row player and column player, respectively. We construct an instance of strategic classification with endogenous costs as follows. The population domain $X = \{\boldsymbol{x}_1, \ldots, \boldsymbol{x}_n\}$ comprises $n$ distinct points. The underlying distribution $\mathcal{D}$ is assumed to be uniform over $X$. The target classifier is $h(\boldsymbol{x}) = 0$ for all $\boldsymbol{x} \in X$.

The Jury chooses a classifier from the set $\{f_1, \ldots, f_m\}$. We associate each classifier $f_j$ with the $j$th column of the game matrices. We define the classifier's outputs to have the opposite label from the column player's utility matrix: $f_j(\boldsymbol{x}_i) = 1 - \mathbf{B}_{ij}$ for all $\boldsymbol{x}_i \in X$. Moreover, because costs are endogenous, the Jury's choice of strategy $j$ also induces a specific cost function $c_j$.

The Contestant chooses a deviation from the set $\{\Delta_1, \ldots, \Delta_n\}$. We restrict these to be *constant deviations*, where $\Delta_i$ maps every input point to the specific point $\boldsymbol{x}_i$ corresponding to the $i$th row. Now, the payoff to the Jury is $\Pr_{\boldsymbol{x} \sim \mathcal{D}}[h(\boldsymbol{x}) = f(\Delta(\boldsymbol{x}))]$. As a result, under a classifier $f_j$ and a deviation $\Delta_i$, the utility of the Jury reads

$$\Pr_{\boldsymbol{x} \sim \mathcal{D}}[h(\boldsymbol{x}) = f_j(\Delta_i(\boldsymbol{x}))] = \Pr_{\boldsymbol{x} \sim \mathcal{D}}[f_j(\Delta_i(\boldsymbol{x})) = 0] = \mathbb{1}[f_j(\boldsymbol{x}_i) = 0] = \mathbf{B}_{ij},$$

so this matches the utility of the column player in the original game. To ensure the Contestant (approximately) maximizes $\mathbf{A}$, we consider the following star metric for each $j$. Each point $\boldsymbol{x}_i$ is connected to a point $\boldsymbol{x}^*$. The cost to go from $\boldsymbol{x}_i$ to $\boldsymbol{x}^*$ is defined as $2M - M\mathbf{A}_{ij}$ for a large parameter $M \gg 1$. Thus, $c_j(\boldsymbol{x}_i, \boldsymbol{x}_{i'}) = c_j(\boldsymbol{x}_i, \boldsymbol{x}^*) + c_j(\boldsymbol{x}_{i'}, \boldsymbol{x}^*)$.

Under a classifier $f_j$ and a deviation $\Delta_i$, the payoff to the Contestant is

$$\mathbb{E}_{\boldsymbol{x} \sim \mathcal{D}}[f_j(\Delta_i(\boldsymbol{x})) - c_j(\boldsymbol{x}, \Delta_i(\boldsymbol{x}))] = f_j(\boldsymbol{x}_i) - \frac{1}{n} \sum_{i'=1}^{n} c_j(\boldsymbol{x}_i, \boldsymbol{x}_{i'}) = f_j(\boldsymbol{x}_i) - \frac{1}{n} \sum_{i'=1}^{n} c_j(\boldsymbol{x}_{i'}, \boldsymbol{x}^*) - c_j(\boldsymbol{x}_i, \boldsymbol{x}^*).$$

The second term above does not depend on the deviation $\Delta_i$, so it is strategically irrelevant. Specifically, we end up with the two-player game with utilities $\langle \boldsymbol{x}, \mathbf{A}'\boldsymbol{y} \rangle + \langle \boldsymbol{c}, \boldsymbol{y} \rangle$ and $\langle \boldsymbol{x}, \mathbf{B}\boldsymbol{y} \rangle$, where $\boldsymbol{c} = (-\frac{1}{n} \sum_{i'=1}^{n} c_j(\boldsymbol{x}_{i'}, \boldsymbol{x}^*))_{j=1}^{m}$, and $\mathbf{A}' = \mathbf{F} + M\mathbf{A}_{ij} - 2M\mathbf{1}$ for $\mathbf{F}_{ij} = f_j(\boldsymbol{x}_i)$; $\mathbf{1}$ denotes the all-ones matrix. Let $(\boldsymbol{x}, \boldsymbol{y}) \in \Delta^n \times \Delta^m$ be Nash equilibrium of this game, which corresponds to a perfomatively stable point of the strategic classification instance. We have $\langle \boldsymbol{x}, \mathbf{A}'\boldsymbol{y} \rangle \geq \langle \hat{\boldsymbol{x}}, \mathbf{A}'\boldsymbol{y} \rangle$ for any $\hat{\boldsymbol{x}} \in \Delta^n$, which implies $\langle \boldsymbol{x}, \mathbf{F}\boldsymbol{y} \rangle + M\langle \boldsymbol{x}, \mathbf{A}\boldsymbol{y} \rangle \geq \langle \hat{\boldsymbol{x}}, \mathbf{F}\boldsymbol{y} \rangle + M\langle \hat{\boldsymbol{x}}, \mathbf{A}\boldsymbol{y} \rangle$. Since $\mathbf{F}_{ij} \in \{0, 1\}$, it follows that $\langle \boldsymbol{x}, \mathbf{A}\boldsymbol{y} \rangle \geq \langle \hat{\boldsymbol{x}}, \mathbf{A}\boldsymbol{y} \rangle - \frac{1}{M}$ for any $\hat{\boldsymbol{x}} \in \Delta^n$. As a result, $(\boldsymbol{x}, \boldsymbol{y})$ is a $1/M$-Nash equilibrium of the original two-player game. $\square$

