# OpenReview forum: "On the Computational Complexity of Performative Prediction"
_ICML.cc/2026/Conference — ICML 2026 regular_

### Official Review · Reviewer_iuNa · 2026-03-10

**Soundness:** 3
**Presentation:** 4
**Significance:** 4
**Originality:** 4
**Overall Recommendation:** 4
**Confidence:** 3

**Summary:**

This paper provides a complete characterization of the computational complexity of finding *Performatively Stable Points*. The central result is the discovery of a sharp computational phase transition: computing an $\epsilon$-performatively stable point is **PPAD-complete** (and thus as hard as finding a Nash equilibrium) when the parameter $\rho = L\beta/\alpha$ exceeds 1, even by an arbitrarily small amount ($\rho = 1 + O(\epsilon)$). This intractability holds even for simple quadratic loss functions and affine distribution shifts. The paper maps the full complexity landscape: it establishes tractability (via linear convergence of Repeated Risk Minimization) for $\rho < 1$, presents a poly$(d, \log(1/\epsilon))$-time algorithm for the marginally expansive regime $\rho \leq 1 + O_{\epsilon}(\epsilon^4)$ (improving upon RRM), and proves unconditional, information-theoretic lower bounds showing that finding a stable point requires exponentially many Empirical Risk Minimization (ERM) evaluations. Key technical extensions include generalizing the hardness result to *general convex constraint sets* (beyond the hypercube) and proving that finding a *local* optimum in strategic classification (a special case) is **PLS-hard**.

**Compliance With Llm Reviewing Policy:**

Affirmed.

**Key Questions For Authors:**

1. The parameter $\rho = L\beta/\alpha$ is central to the phase transition. Could you provide more discussion or intuition on the practical prevalence of the $\rho > 1$ regime? In real-world applications of performative prediction (e.g., strategic classification, recommendation systems), are there empirical studies or theoretical reasons to believe that $\rho$ often exceeds 1, or is this primarily a worst-case hardness result? Clarifying the practical implications of your hardness theorems would help assess their immediate impact.

2. Theorem 3.5 presents a poly$(d, \log(1/\epsilon))$-time algorithm for the regime $\rho \leq 1 + O_{\epsilon}(\epsilon^4)$, based on the ellipsoid method applied to a (hypomonotone) Variational Inequality. While its theoretical runtime is attractive, what are the practical computational bottlenecks of this algorithm, particularly the cost of implementing the required separation oracle for the mapping $F(x)=x-G(x)$? How does this cost compare, in practical terms, to a single iteration of simple iterative methods like RRM? A brief discussion on its practical feasibility versus its theoretical contribution would be valuable.

3. The reduction to general convex domains (Theorem 3.12, Corollary 3.13) requires the domain to be "well-bounded" and to contain a 2D ball. Many common constraint sets in machine learning (e.g., the probability simplex, the $\ell_1$-ball) are convex and compact but are lower-dimensional. Does your hardness result extend to such lower-dimensional convex sets? If not, this is an important limitation to clarify, as it affects the generality of the claim.

4. The paper shows that finding a local optimum in strategic classification is PLS-hard. While the reduction is from LOCALMAXCUT, the result itself doesn't directly analyze specific heuristics. Does this PLS-hardness result provide any new insight into the performance of specific local search heuristics commonly used in practice (e.g., best-response dynamics, gradient-based methods)? Could you speculate on whether this hardness is more a theoretical artifact or if it hints at potential convergence difficulties for such heuristics in practice?

**Limitations:**

The authors discuss some limitations and open questions in the conclusion (e.g., the gap in Figure 1, the complexity of performative stability in standard strategic classification). A standard "Impact Statement" is included. However, the discussion of limitations could be strengthened. Specifically, the paper would benefit from a more explicit discussion of: 1) The worst-case nature of the hardness results; the paper does not address whether the problem might be easier on average or for instances arising from specific applications. 2) The dependence on the standard performative prediction model; as noted in the related work (Section 1.2), there is research on misspecification (Xue \& Sun, 2024), but the paper does not discuss the robustness of its complexity conclusions to such model deviations. 3) While extending to general norms, the connection between the abstract contraction condition on the RRM map $G$ and the fundamental parameters $L, \beta, \alpha$ could be clearer, especially in the non-Euclidean setting (Proposition 3.10).

**Strengths And Weaknesses:**

1. **Soundness:**The paper is technically excellent. The core hardness results are built via meticulous, canonical reductions from well-established PPAD-complete and PLS-complete problems (variational inequalities, fixed points, LOCALMAXCUT) to the performative stability problem. The proofs, detailed in the appendix, appear rigorous and correct. The assumptions (strong convexity, smoothness, L-sensitivity) are standard for the theoretical analysis. The paper honestly presents a balanced picture, discussing both tractable algorithms (e.g., the ellipsoid method for $\rho \leq 1+O_{\epsilon}(\epsilon^4)$, and the recent result of Diakonikolas (2025) for $\rho \leq 1+O_{\epsilon}(\epsilon)$) and intractability results, culminating in a clear complexity landscape (Figure 1). As a purely theoretical work, no experimental evaluation is required.

2. **Presentation:** The presentation is clear and well-structured. The paper logically moves from motivation and definitions to main theorems, extensions, and conclusions. The narrative is coherent. However, the work is *inherently dense* due to its deep engagement with computational complexity theory (PPAD, PLS, reductions). For readers not specialized in this subfield, the heavy reliance on these concepts and the technical notation presents a significant barrier to accessibility. While the related work is comprehensive, the paper could improve accessibility by providing a high-level, intuitive roadmap of the proof strategies and the broader significance of the complexity classes for a general machine learning audience. A dedicated "technical overview" section would be beneficial.

3. **Significance:** The significance of this work is high. It answers a foundational question in the important area of Performative Prediction: is the failure of simple retraining algorithms (like RRM) when $\rho \ge 1$ an algorithmic limitation or an intrinsic computational barrier? The paper provides a definitive answer—it is an intrinsic, PPAD-hard barrier. This fundamentally advances our understanding of the limits of efficient computation in systems with strategic feedback. It will likely steer future research away from seeking universally efficient algorithms for stable points in the expansive regime and towards exploring special cases, heuristics, or different solution concepts. The extension to general convex domains is also a notable standalone contribution to the complexity theory of variational inequalities.

4. **Originality:** The originality is high. While the technical tools (reductions from VI/fixed points) are established in complexity theory, the *novel, systematic, and rigorous application* of these tools to completely characterize the complexity of Performative Prediction is a clear and important contribution. The discovery of a sharp computational phase transition at $\rho = 1$ is a novel and insightful theoretical finding. The paper does not merely apply an existing reduction; it carefully tailors constructions to the specific structure of performative prediction and pushes boundaries by handling general convex constraint sets. The PLS-hardness result for local strategic optimality is another original contribution that deepens the understanding of hardness in this domain.

---

> ### Author Rebuttal · Authors · 2026-03-31
>
> We thank the reviewer for their service and valuable feedback.
>
> > "The parameter $\rho = L\beta/\alpha$ is central to the phase transition. Could you provide more discussion or intuition on the practical prevalence of the $\rho > 1$ regime? In real-world applications of performative prediction (e.g., strategic classification, recommendation systems), are there empirical studies or theoretical reasons to believe that $\rho$ often exceeds 1, or is this primarily a worst-case hardness result? Clarifying the practical implications of your hardness theorems would help assess their immediate impact."
>
> The condition $\rho < 1$ is quite restrictive, as it requires strong assumptions like strong convexity. If the loss function is not strongly convex, then $\rho$ is in fact unbounded. In many practical applications the loss function is not even convex, let alone strongly convex (for example, we refer to ``Stochastic Optimization Schemes for Performative Prediction with Nonconvex Loss'' by Li and Wai, 2025). Indeed, the regime where $\rho > 1$ is very common and arises even in simple problems, as noted first by Perdomo et al. (2020) in their experimental simulations. Overall, given the complexity of modern ML systems, we do not expect the condition $\rho < 1$ to be satisfied, which motivates understanding the regime where $\rho > 1$. We will include this discussion in the revised version.
>
> > "Theorem 3.5 presents a $poly(d, \log (1/\epsilon))$-time algorithm for the regime $\rho > 1 + O_{\epsilon} (\epsilon^4)$, based on the ellipsoid method applied to a (hypomonotone) Variational Inequality. While its theoretical runtime is attractive, what are the practical computational bottlenecks of this algorithm, particularly the cost of implementing the required separation oracle for the mapping $F(x) = x - G(x)$? How does this cost compare, in practical terms, to a single iteration of simple iterative methods like RRM?"
>
> We thank the reviewer for raising this point. While the ellipsoid algorithm has a running time polynomial in $\log(1/\epsilon)$, making it very attractive in theory, it is not the most practical algorithm. As we point out in Corollary 3.9, a more practical algorithm is based on Halpern's iteration. Its cost is comparable to simple iterative methods such as RRM, and at the same time succeeds even when $\rho = 1 + O(\epsilon)$ (in contrast, RRM can fail even when $\rho = 1$.)
>
> > "The reduction to general convex domains (Theorem 3.12, Corollary 3.13) requires the domain to be "well-bounded" and to contain a 2D ball. Many common constraint sets in machine learning (e.g., the probability simplex, the $\ell_1$ ball) are convex and compact but are lower-dimensional. Does your hardness result extend to such lower-dimensional convex sets?"
>
> We clarify that the well-boundedness assumption only requires the domain to contain a two-dimensional ball (see Lines 309 - 312). This condition is typically satisfied even for low-dimensional domains (provided the dimension is at least two). In particular, the probability simplex, the $\ell_1$ ball, and general convex polytopes all satisfy this assumption. Furthermore, this assumption is standard in optimization theory.
>
> > "The paper shows that finding a local optimum in strategic classification is PLS-hard. While the reduction is from LOCALMAXCUT, the result itself doesn't directly analyze specific heuristics. Does this PLS-hardness result provide any new insight into the performance of specific local search heuristics commonly used in practice (e.g., best-response dynamics, gradient-based methods)? Could you speculate on whether this hardness is more a theoretical artifact or if it hints at potential convergence difficulties for such heuristics in practice?"
>
> Indeed, our reduction shows that no local search heuristic, such as algorithms based on best-response dynamics, can converge in polynomial time in the worst case (under the well-believed complexity assumption PLS $\neq$ P). A compelling aspect of our approach based on complexity theory is that it rules out *any* local search algorithm, potentially much more sophisticated than best-response dynamics.
>
> > "However, the discussion of limitations could be strengthened. Specifically, ..."
>
> We thank the reviewer for bringing up these points, we will make sure to incorporate them. In the revised version, we will highlight further that it is plausible that one can bypass our hardness result in more structured instances. Concerning misspecification, our main goal in the paper was to prove hardness results; misspecification can only make the problem harder. It is a strength of our hardness result that it applies even when there is no misspecification. We are also happy to adapt the statement of Proposition 3.10 to make it more clear, as the reviewer sees fit.

---

### Official Review · Reviewer_iar5 · 2026-03-11

**Soundness:** 3
**Presentation:** 3
**Significance:** 4
**Originality:** 3
**Overall Recommendation:** 4
**Confidence:** 3

**Summary:**

This paper takes on the problem of characterizing the computational complexity of prediction under performativity, i.e., when the decision you take or the model you deploy influence (causes a shift) your data distribution. For example, in loan-lending by banks, or recommendation systems etc. Discussions through the paper revolves around the regimes when the performative strength of the environment is low $(\rho < 1)$ and high $(\rho \geq 1)$. The results show, in case when $\rho \leq 1 + \mathcal{O}(\epsilon^4)$, computing an $\epsilon$-stable point solution is *polynomial* time solvable. On the other hand, when $\rho > 1 + \mathcal{O}(\epsilon)$, the same problem is *PPAD-complete*, which means the problem is hard as achieving Nash equilibrium in general sum games. The authors argue that the hardness of the problem does not lie in the complexity of the algorithm we use, instead the tractable region is too narrow under performativity. Even with simple modelling assumptions like quadratic loss, gradual shifting in environments, and convex parameter space, the problem still remains hard to solve, and the complexity measures persists.

**Compliance With Llm Reviewing Policy:**

Affirmed.

**Final Justification:**

I have been positive regarding this work since the beginning. I believe it addresses a critical lacunae the current performative learning literature. The theoretical results on hardness are interesting. The rebuttal was crisp, but detailed. It indeed reinforce my initial evaluation. Thus, I advocate for a positive evaluation for this work.

**Key Questions For Authors:**

- **Related works:** The related work in this paper misses important literature on performative learning in Sequential decision making. For example, [1,2] uses RRM training method to reach performative stability. Recently, [3] shows $\epsilon$-performative optimality is actually reachable for performative MDPs. The paper could discuss more connections to work studying stateful environments, such as [4].

- The authors does not discuss anything on complexity achieving performative optimality. How this analysis translate in finding performative optimality in Offline ML?

- The positive result in this paper used ellipsoid algorithm. In practice, this algorithm can be often very slow. In real pipelines, does the results still hold for usable (fast) algorithms?

- As per my understanding, the authors assume they have access to an ERM oracle. In learning theory, this assumption seems fair, but as this paper focuses on complexity of the problem, we should consider that training itself is expensive. In practical ML pipelines, we often have access to an approximate oracle. How does your complexity results addresses this gap?

-  I understand this paper is purely theoretical, though some implementations on real ML pipelines would compliment the theoretical guarantees. Additionally, the authors should discuss how large $\rho$ can be, or does real ML systems lie mostly in hard phase $(\rho >1)$?

- **Additional question.** Can the authors comment anything on the analysis if we do not have access to the true data distribution $\mathcal{D}(\theta)$?

- **Suggestion on Introduction:** The initial discussion in the introduction heavily relies on notations such as $L,\alpha,\beta$. As a reader, I felt a bit disconnected from a generic motivation of the paper. I would suggest a revision on the introduction appending a general discussion on failure of RRM methods and then going into mathematical notations.


**References:**
[1] Mandal, Debmalya, Stelios Triantafyllou, and Goran Radanovic. "Performative reinforcement learning." International Conference on Machine Learning. PMLR, 2023.
[2] Rank, Ben, et al. "Performative Reinforcement Learning in Gradually Shifting Environments." Uncertainty in Artificial Intelligence. PMLR, 2024.
[3] Basu, Debabrota, et al. "Performative Policy Gradient: Optimality in Performative Reinforcement Learning." arXiv preprint arXiv:2512.20576 (2025).
[4] Brown, Gavin, Shlomi Hod, and Iden Kalemaj. "Performative prediction in a stateful world." International conference on artificial intelligence and statistics. PMLR, 2022.

**Limitations:**

Yes

**Strengths And Weaknesses:**

- **Soundness:** The paper is technically sound.
- **Presented:** The paper is well-presented. It is easy to parse through. Though I have some suggestions regarding the introduction and related works. See Questions.
- **Significance:** The problem addressed in this paper, in my opinion is very important. As performative prediction has gained more and more attention in recent years, it is imperative to understand it's complexity and phase transitions based on the performative strength ($\rho$) of the environment. Moreover this work bridges a gap among topics like game theory, complexity theory and as a whole local optimization.
- **Originality:** To best of my knowledge, no prior work focuses solely on the computational complexity of achieving equilibrium under shifting targets in context of performative prediction in machine learning. As performative stability is being increasingly explored in offline ML and sequential ML, this paper can be placed very naturally in the existing literature.

---

> ### Author Rebuttal · Authors · 2026-03-30
>
> We thank the reviewer for their service and valuable feedback.
>
> > "Related works: The related work in this paper misses important literature on performative learning in Sequential decision making. For example..."
>
> We thank the reviewer for bringing those papers to our attention. We will make sure to cite and discuss them in the revised version of our paper, as they are important in this line of work.
>
> > "The authors does not discuss anything on complexity achieving performative optimality. How this analysis translate in finding performative optimality in Offline ML?"
>
> We thank the reviewer for raising this point. We point out that finding a performative optimum is NP-hard via a straightforward reduction from nonconvex optimization. Specifically, consider the loss function $\ell(x, z) = z$, and let the distribution $\mathcal{D}(x)$ to be a singleton such that $z = g(x)$ for an arbitrary nonconvex function $g(\cdot).$ In this setting, finding a performative optimum is equivalent to finding the global minimum of $g(x)$. We will include this remark in the revised version.
>
> It is also worth noting that in the strategic classification setting (which can be viewed as a discrete analogue of performative prediction), finding a strategically optimal solution has been shown to be NP-hard (Hardt et al., 2016).
>
> > "The positive result in this paper used ellipsoid algorithm. In practice, this algorithm can be often very slow. In real pipelines, does the results still hold for usable (fast) algorithms?"
>
> While we used the ellipsoid algorithm to obtain a running time polynomial in $\log(1/\epsilon)$, a more practical alternative is to use, for example, Halpern's iteration (see Diakonikolas, 2025); this is pointed out in Corollary 3.9. While the complexity of Corollary 3.9 is linear in $1/\epsilon$, which is in theory inferior relative to the ellipsoid in terms of $1/\epsilon$, Halpern's iteration is very much a practical algorithm and enjoys a dimension-free convergence bound (Diakonikolas, 2025).
>
> > "As per my understanding, the authors assume they have access to an ERM oracle. In learning theory, this assumption seems fair, but as this paper focuses on complexity of the problem, we should consider that training itself is expensive. In practical ML pipelines, we often have access to an approximate oracle. How does your complexity results addresses this gap? / Can the authors comment anything on the analysis if we do not have access to the true data distribution?"
>
> We clarify that our complexity results hold under an approximate oracle or if we do not have access to the true data distribution. Since we establish a lower bound for computing a performatively stable point, assuming access to an approximate oracle or samples from the induced distribution would only make the computational problem more challenging, so our hardness result continues to hold in such settings. We will include this clarification in the revision.
>
> > "I understand this paper is purely theoretical, though some implementations on real ML pipelines would compliment the theoretical guarantees. Additionally, the authors should discuss how large can be, or does real ML systems lie mostly in hard phase $(\rho > 1)$ ?"
>
> While our paper has a theoretical scope, we are primarily motivated by the extensive prior work that has shown that simple heuristics such as RRM often fail to converge in practical instances. Our aim is to explain from a theoretical standpoint the behavior previously observed in practical instances.
>
> Regarding the magnitude of $\rho$, the condition $\rho < 1$ is quite restrictive, as it requires strong assumptions like strong convexity. If the loss function is not strongly convex, then $\rho$ is in fact unbounded. In many practical applications the loss function is not even convex, let alone strongly convex (for example, we refer to "Stochastic Optimization Schemes for Performative Prediction with Nonconvex Loss'' by Li and Wai, 2025). Indeed, the regime where $\rho > 1$ is very common and arises even in simple problems, as noted first by Perdomo et al. (2020) in their experimental simulations. Overall, given the complexity of modern ML systems, we do not expect the condition $\rho < 1$ to be satisfied, which motivates understanding the regime where $\rho > 1$. We will include this discussion in the revised version.
>
> > Suggestion on Introduction: ... I would suggest a revision on the introduction appending a general discussion on failure of RRM methods and then going into mathematical notations."
>
> We thank the reviewer for the suggestion. We see how our introduction is somewhat notation heavy. We will make sure to incorporate the reviewer's suggestion in the revised version.

---

> > ### Author Rebuttal · Reviewer_iar5 · 2026-04-03
> >
> > I thank the authors for their response. I remain positive about this work.

---

### Official Review · Reviewer_iqoX · 2026-03-15

**Soundness:** 4
**Presentation:** 4
**Significance:** 2
**Originality:** 3
**Overall Recommendation:** 5
**Confidence:** 3

**Summary:**

This paper studies the computational hardness of the performance stability in the performative prediction problem. The main result of the paper is that computing an $\varepsilon$-performatively stable point is PPAD-hard even when $\rho \leq 1 + O(\varepsilon)$. The paper also gives a tractability result in a narrower near-threshold regime and additional results for general domains and strategic classification.

**Compliance With Llm Reviewing Policy:**

Affirmed.

**Key Questions For Authors:**

I don't have a specific technical question. However, for completeness, it would add more value if the authors could elaborate on their statement in conclusion section regarding the research gap. More specifically, how should readers interpret this gap: is it mainly a technical limitation of current proof methods, or is there evidence that the exact transition may be more subtle?

**Limitations:**

Yes

**Strengths And Weaknesses:**

Strengths: The paper is very well written and well structured. The topic studied in this paper is relevant to the conference. The paper's contributions are original and novel. While some of the reduction techniques used in the paper are well-known, the paper cleverly applies them to prove the problem. The paper clearly closes the research gap in computational complexity issues in performative prediction.

Weaknesses: My main concern is significance. The paper clearly establishes a computational complexity regime, which is valuable, but the contribution is mostly classificatory. The paper offers limited new structural and algorithmic insights into performative prediction.  The conclusion also makes clear that important gaps remain between the current tractability result, recent related algorithmic work, and the hardness threshold.

---

> ### Author Rebuttal · Authors · 2026-03-30
>
> We thank the reviewer for their valuable time and positive comments.
>
> > "My main concern is significance. The paper clearly establishes a computational complexity regime, which is valuable, but the contribution is mostly classificatory. The paper offers limited new structural and algorithmic insights into performative prediction."
>
> We believe that our complexity classification does provide important algorithmic and structural insights. It explains why certain standard algorithmic approaches, such as RRM, can fail or take exponential time. More than that, it shows that *any* new algorithm, potentially much more sophisticated, will face exactly the same limitations. As a result, our complexity threshold will guide future algorithm design, motivating the need for tractable relaxations of performative stability or additional structural assumptions. For example, while preparing this rebuttal, we came across a preprint by Farina and Perdomo ("The Stability of Online Algorithms in Performative Prediction") that introduced a relaxation of performative stability, directly motivated by our hardness result.
>
> > "The conclusion also makes clear that important gaps remain between the current tractability result, recent related algorithmic work, and the hardness threshold."
>
> We clarify that in the regime where $\epsilon$ is inverse polynomial, which is more relevant in practice, our hardness result matches (up to constants) the recent upper bound of Diakonikolas (2025). We agree with the reviewer in that there are still certain gaps in other regimes (up to constant factors in $\epsilon$), but we consider that to be a strength of the paper, as it sets up a concrete research agenda for future work.
>
> > "However, for completeness, it would add more value if the authors could elaborate on their statement in conclusion section regarding the research gap. More specifically, how should readers interpret this gap: is it mainly a technical limitation of current proof methods, or is there evidence that the exact transition may be more subtle?"
>
> We thank the reviewer for raising this question. It is not clear to us whether the gap is a technical limitation or there is a fundamental reason for it. This complexity question was entirely unexplored prior to our paper, so there is not much evidence in either direction. We believe that this is an interesting direction for future work.

---

> > ### Author Rebuttal · Reviewer_iqoX · 2026-04-03
> >
> > I thank the authors for feedback. The feedback adequately addresses all the concerns raised.

---

### Decision · Program_Chairs · 2026-04-30

**Decision:**

Accept (regular)

**Comment:**

This paper investigates performative prediction, a setting where model predictions influence the underlying data distribution (e.g., a bank's default risk model affecting actual default behavior). The authors characterize the computational complexity of finding Performatively Stable Points, which are equilibrium states where predictions align with the induced data. A sharp phase transition is identified: the problem is tractable when the performative strength parameter \gamma <= 1, but becomes PPAD-hard when \gamma > 1. This result is both novel and theoretically significant. While the proposed ellipsoid-based algorithm for the threshold case may be impractical due to its runtime, it successfully establishes a clean theoretical boundary for tractability. Overall, this paper represents a strong and well-rounded theoretical contribution to the field.